# Patient-reported outcomes predict return to work and health-related quality of life six months after cardiac rehabilitation: Results from a German multi-centre registry (OutCaRe)

**Annett Salzwedel**[1]*, **Iryna Koran**[2], **Eike Langheim**[3], **Axel Schlitt**[4], **Jörg Nothroff**[5], **Christa Bongarth**[6], **Markus Wrenger**[7], **Susanne Sehner**[8], **Rona Reibis**[9], **Karl Wegscheider**[8], **Heinz Völler**[1,2], **for the OutCaRe investigators**¶

1 Department of Rehabilitation Medicine, Faculty of Health Sciences Brandenburg, University of Potsdam, Potsdam, Germany, 2 Klinik am See, Rehabilitation Centre of Cardiovascular Diseases, Rüdersdorf, Germany, 3 Reha-Zentrum Seehof der Deutschen Rentenversicherung Bund, Teltow, Germany, 4 Paracelsus-Klinik Bad Suderode, Quedlinburg, Germany, 5 MediClin Reha-Zentrum Spreewald, Burg, Germany, 6 Klinik Höhenried, Bernried, Germany, 7 Caspar Heinrich Klinik, Bad Driburg, Germany, 8 Institute for Medical Biometry and Epidemiology, University Medical Center Hamburg-Eppendorf, Hamburg, Germany, 9 Cardiological Outpatient Clinic am Park Sanssouci, Potsdam, Germany

¶ Membership of the OutCaRe investigators is listed in the Acknowledgments.
* annett.salzwedel@fgw-brandenburg.de

**Data Availability Statement:** Data are available (https://zenodo.org/badge/DOI/10.5281/zenodo.

## Abstract

### Background

Multi-component cardiac rehabilitation (CR) is performed to achieve an improved prognosis, superior health-related quality of life (HRQL) and occupational resumption through the management of cardiovascular risk factors, as well as improvement of physical performance and patients' subjective health. Out of a multitude of variables gathered at CR admission and discharge, we aimed to identify predictors of returning to work (RTW) and HRQL 6 months after CR.

### Design

Prospective observational multi-centre study, enrolment in CR between 05/2017 and 05/2018.

### Method

Besides general data (e.g. age, sex, diagnoses), parameters of risk factor management (e.g. smoking, hypertension), physical performance (e.g. maximum exercise capacity, endurance training load, 6-min walking distance) and patient-reported outcome measures (e.g. depression, anxiety, HRQL, subjective well-being, somatic and mental health, pain, lifestyle change motivation, general self-efficacy, pension desire and self-assessment of the occupational prognosis using several questionnaires) were documented at CR admission

3605618.svg) and citable as Salzwedel, Annett, Balzer, Klaus, Koran, Iryna, Langheim, Eike, Schlitt, Axel, Nothroff, Axel, . . . Völler, Heinz. (2020). Patient-reported outcomes predict return to work and health-related quality of life 6 months after cardiac rehabilitation: Results from a German multi-centre registry (OutCaRe): dataset [Data set]. Zenodo. http://doi.org/10.5281/zenodo.3605618.

**Funding:** This work was supported by the German Federal Pension Insurance (grant No. 8011-106-31/31.114.1). Funding was granted to the University of Potsdam. URL: https://www.deutsche-rentenversicherung.de/DRV/DE/Experten/Reha-Forschung/Forschungsfoerderung/Forschungsfoerderung.html The funder had no role in study design, data collection and analysis, decision to publish, or preparation of the manuscript.

**Competing interests:** The authors have declared that no competing interests exist.

and discharge. These variables (at both measurement times and as changes during CR) were analysed using multiple linear regression models regarding their predictive value for RTW status and HRQL (SF-12) six months after CR.

## Results

Out of 1262 patients (54±7 years, 77% men), 864 patients (69%) returned to work. Predictors of failed RTW were primarily the desire to receive pension (OR = 0.33, 95% CI: 0.22–0.50) and negative self-assessed occupational prognosis (OR = 0.34, 95% CI: 0.24–0.48) at CR discharge, acute coronary syndrome (OR = 0.64, 95% CI: 0.47–0.88) and comorbid heart failure (OR = 0.51, 95% CI: 0.30–0.87). High educational level, stress at work and physical and mental HRQL were associated with successful RTW. HRQL was determined predominantly by patient-reported outcome measures (e.g. pension desire, self-assessed health prognosis, anxiety, physical/mental HRQL/health, stress, well-being and self-efficacy) rather than by clinical parameters or physical performance.

## Conclusion

Patient-reported outcome measures predominantly influenced return to work and HRQL in patients with heart disease. Therefore, the multi-component CR approach focussing on psychosocial support is crucial for subjective health prognosis and occupational resumption.

## Trial registration

The study was registered at the German Clinical Trial Registry and the International Clinical Trials Registry Platform (ICTRP) of the World Health Organization (DRKS00011418; http://www.drks.de/DRKS00011418, http://apps.who.int/trialsearch/Trial2.aspx?TrialID=DRKS00011418).

## Introduction

According to recommendations by the European Association of Preventive Cardiology, cardiac rehabilitation (CR) should include medical examination, exercise training, cardiovascular risk factor management (e.g. lipid and blood pressure monitoring, smoking cessation), lifestyle counselling in terms of physical activity and nutrition, as well as vocational support and psychosocial management. [1] Through this multimodal approach, an improved clinical prognosis, high quality of life and social participation should be achieved. [1]

While the positive effect of CR on the clinical course and health-related quality of life (HRQL) has been sufficiently proven, [2–6] the evidence regarding the impact on occupational outcomes and return to work rates remains inconclusive. [7] However, CR programmes based on a multi-modal approach seem to be at an advantage over exercise-based programmes in terms of both clinical and occupational outcomes. [3,7] Nevertheless, a significant proportion of patients (30% of patients in Germany) [8] of employable age fail to return to work after CR, resulting in a psychosocial burden on the patient with reduced HRQL [9] and high costs to society. To overcome this gap, we need optimised CR programmes adapted to the individual needs of patients. Knowledge of modifiable parameters predicting return to work and HRQL, which reflect the holistic interprofessional therapy options of the multi-modal CR, could

provide important clues. In this regard, patient-reported outcome measures must be considered, as they have become increasingly important as a measure of patient-centred care and as an integral part of high-quality healthcare over the last decade. [10,11]

In general, return to work of patients with cardiovascular disease depends on several clinical and contextual factors, such as the patient's physical performance, depression, anxiety, expectations of professional reintegration and perceived health, with psychosocial parameters appearing more influential than classical cardiovascular risk factors. [12–16] Even for HRQL in patients living with heart disease, influencing factors such as exercise, coronary artery bypass surgery, socioeconomic parameters and a poor physician-patient relationship have been investigated. [17–19] Since return to work and HRQL after a cardiac event are closely associated, [15] several common predictors can be assumed.

However, to date there are no systematic and prospective research data on the associations between a multitude of modifiable parameters presenting the most important core components of multi-modal CR and return to work, as well as HRQL over the mid-term course (three months to one year) after CR discharge. In the Outcome of Cardiac Rehabilitation (OutCaRe) study, we aimed to identify predictors of occupational reintegration and HRQL among the same set of patient-reported outcome measures, clinical parameters, cardiovascular risk factors and physical performance.

We hypothesised that modifications of patient-reported outcome measures during CR independent of cardiovascular risk factors and physical performance would predict return to work, as well as HRQL six months after CR.

## Methods

### Study design

The OutCaRe study was designed to identify and evaluate performance measures of CR. The study was conducted following a stepwise concept. First, a Delphi expert survey of 70 cardiologists, psychologists, sports therapists and physiotherapists extracted potential indicators of rehabilitation success, including parameters of the four key areas of CR: cardiovascular risk factors; physical performance; social medicine and subjective health. [20]

Subsequently, a national multi-centre register study was performed to evaluate the predefined indicators regarding feasibility and modifiability of the CR routine. [21] This paper analyses these parameters in terms of their mid-term predictive value for return to work and HRQL after CR.

### Patients and cardiac rehabilitation

In 12 German rehabilitation centres, eligible patients aged up to 65 years were enrolled at admission to CR, regardless of their primary allocation diagnoses (e.g. acute myocardial infarction, coronary artery bypass grafting, coronary artery disease, valvular disease, vascular disease, implantation of active electrical devices (implantable cardioverter defibrillator or cardiac resynchronisation therapy)). Patients with insufficient German language skills, early retirement or missing consent were excluded.

All patients performed a standardised comprehensive CR programme with a regular duration of three to four weeks according to the specifications of the German pension insurance. [21] The programme could be performed in either an inpatient or outpatient setting and consisted of all-day activities including counseling by a cardiologist, risk-factor modification strategies (e.g. patient education on nutritional habits, smoking cessation, physical activity and medication adherence), physician-supervised exercise training and sports therapy (e.g. training on a bicycle ergometer, outdoor walking, resistance training, gymnastics), psychosocial

interventions (health education and counseling, psychotherapy, stress management in single or group sessions), and vocational assessment and physician and social worker counseling. [22,23] On average, patients performed 12 training and sports therapy units per week with a duration, depending on the training group and physical performance, of up to 30 minutes and 45 minutes for outdoor walking, respectively, and 8 additional counselling sessions. [24]

## Predictors, data capturing

The data capture was realised via a web-based electronic case report form provided by Secu-Trial® (interActive Systems, Berlin).

Upon admission to CR, sociodemographic data (e.g. age, sex, educational level), diagnoses, procedures and relevant comorbidities (diabetes mellitus, musculoskeletal disease, psychological diagnosis) and—as potential confounders of rehabilitation outcomes—perceived occupational stress and incisive life events using the INTERHEART stress scale [25] were recorded. Additionally, modifiable cardiovascular risk factors, parameters of physical performance, social medicine and subjective health operationalised by a multitude of patient-reported performance measures were captured at both CR admission and discharge. The majority of captured data were taken from patients' records, while social medicine and subjective health were assessed by means of specific patient-reported outcome measures used in the study:

- **Cardiovascular risk factors**: smoking behaviour, systolic/diastolic blood pressure, low density lipoprotein (LDL) cholesterol.

- **Physical performance**: maximum exercise capacity on the bicycle stress test, endurance training load in ergometry, 6-min walking distance.

- **Social medicine and subjective health in patient-reported outcome measures**:

  - depression on the patient health questionnaire (PHQ-9), [26]

  - heart-focussed anxiety using the cardiac anxiety questionnaire with the aspects *fear*, *avoidance* and *attention* (revised German version: Herzangstfragebogen, HAF-17), [27]

  - subjective well-being on the World Health Organization questionnaire (WHO-5), [28]

  - quality of life on the short-form health survey (SF-12) with the physical and mental health component summary scales, [29]

  - subjective health using indicators of rehabilitation status (IRES-24) with the subscales *pain*, *somatic health* and *mental health*, [30]

  - general self-efficacy expectations on the short scale for measuring general self-efficacy beliefs (ASKU; range 1–5 points), [31]

  - lifestyle change motivation (single choice question: Can you imagine changing your lifestyle due to your condition? Answer options: definitely; probably; uncertain; probably not; no way),

  - self-assessed health prognosis (single choice question: Please estimate what state of health you expect after the next 6 months. Answer options: excellent; very good; good; less good; poor),

  - pension desire and self-assessed occupational prognosis using the Würzburger Screening identifying occupational issues and the need for vocational rehabilitation. [32]

A more detailed description of parameter operationalisation is published elsewhere. [21]

## Outcome measures

A follow-up survey conducted by mail (paper-pencil procedure) or e-mail with a direct link to the electronic case report platform regarding patients' preferences 6 months after CR, return to work status and HRQL were assessed as outcome measures by the physical and mental component summary scales in the SF-12.

## Ethics approval and study registration

All patients were informed about the contents of the study and provided written informed consent prior to enrolment. The study was approved by the Ethics Committee of the State Medical Association of Brandenburg (approval number S 4(a)/2017) as the responsible institutional review board of the principal investigator (HV, Klinik am See, Rüdersdorf, Germany) and, additionally, by the local institutional review board for each participating CR centre. Out-CaRe is registered with the German Register of Clinical Trials and the International Clinical Trial Agency (ICTRP, WHO; registration number DRKS00011418).

## Statistics

For description, continuous variables are presented as means ± standard deviation, and categorical variables as absolute values and percentages. Differences in variables between admission to CR, discharge from CR and follow-up were tested for statistical significance using Wilcoxon tests for continuous variables and McNemar tests for categorical variables. For metric variables, the standardised effect size (SES) was additionally calculated (ratio of mean value differences and pretest standard deviation), [33] which was interpreted according to Cohen, who defined an effect size of 0.20 as small, 0.50 as moderate and 0.80 or greater as large. [34] In addition, the changes in patient-reported outcome measures (metric variables) during CR were assessed using the minimal important difference (MID) if available. This concerns the IRES-24 (MID 0.5 points) [35] and the SF-12 (MID two points for the physical and three points for the mental component summary). [36] For the WHO-5, we anticipate a MID of 10 percentage -points. [28] No MID was considered for PHQ-9 because the MID reported in the literature focusses on the change of PHQ values in the acute phase of depression treatment in affected patients, [37] which is of subordinate relevance in our investigation. There is no MID available for the HAF-17.

While the descriptive analyses of baseline and rehabilitation data used the original available data without imputation, an imputation model was developed to allow more complex analyses with many predictors despite missing values. Missing values were imputed using chained equations (MICE) with 20 imputations. All psychological scores were imputed by predictive mean matching with the ten nearest neighbours; for all other variables, a parametric approach depending on the measurement level of the variable was chosen. For all further analyses, only patients with a non-missing 6-month outcome in the original dataset were included. Return to work was analysed regardless of the employment status at baseline, as it is a goal of CR in Germany to support return to work even in patients who were on sick leave or unemployed before CR. Predictors of patients' return to work and HRQL were identified using multiple linear regression models with the imputed baseline and discharge data. With this full model as the starting model (see supplemental material, S1–S3 Figs), variable selections were performed as described by Wood et al. [38] A backward selection, using the likelihood-ratio test with a significance level for removal of p = 0.05, was executed with a weighted linear regression for the stacked imputed data set. The weights were calculated with respect to the number of imputations and the average fraction of missing data across all variables. The selected models were fitted for each imputed dataset separately and the parameter estimates were combined using

Rubin's Rule. [39] To control the reason for CR, we simplified the variable to 'acute coronary syndrome' vs 'no acute coronary syndrome'. This variable was selected for the final model (the marginal means of RTW by reason for CR are shown in the supplement, S4 Fig). For the complete set of potential regressors, potential multicollinearity was studied by calculating the variance inflation factors, values below 10 were considered acceptable.

Results are presented as forest plots with 95% confidence intervals (95% CI) and *p*-values. Effects with a *p*-value of less than 0.05 were considered statistically significant. Calculations were carried out using STATA 15.0 and SPSS Version 25.0.

## Results

### Patient characteristics and cardiac rehabilitation

Out of a total of 1,586 enrolled patients, 1,262 participants (80%) responded to the follow-up questionnaires and could be analysed (Fig 1). The majority of these patients (mean age 54±7 years, 77% male) had completed at least 10 years of education (78%), lived in a family or with a partner (79%) and were employed before CR (90%), though 72% were on sick leave. The patients were assigned to CR primarily due to an acute coronary syndrome (40%) or coronary artery disease without myocardial infarction (18%), followed by heart valve diseases in 12% of patients and coronary artery bypass grafting (8%). CR was most often conducted after an acute event (83%) and in an inpatient setting (92% of patients). Patients stayed an average of 23.7 ±4.5 days in CR and most were discharged normally. Eighteen patients (1.4%) were discharged

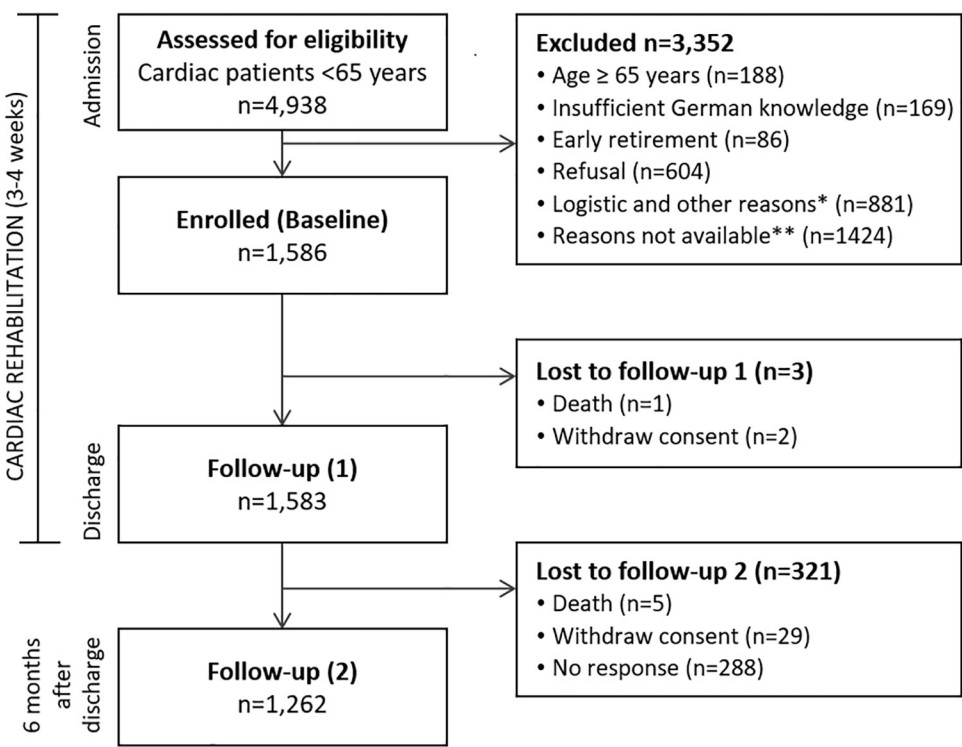

**Fig 1. Flowchart of patient recruitment and study process.** *In CR centres, documented reasons for exclusion were earlier discharge, personal reasons, limitations due to orthopaedic/psychological restrictions, not interested in participation, did not fill in questionnaires, assessment for eligibility too late, no cardiac referral diagnosis, dyslexia. **In 5 CR centres, the reasons for exclusion were not documented.

prematurely at their own request, and nine (0.7%) were transferred to acute hospital care. Patient characteristics, diagnoses, risk factors and comorbidities are presented in Table 1.

During the CR program, several cardiovascular risk factors, physical performance and patient-reported outcome measures were improved. The proportion of current smokers was significantly reduced from 35% to 15%. The mean endurance training load was increased by 21 watts. The percentage of lifestyle change-motivated patients and those who reported excellent or very good self-assessed health prognoses was improved ($p < 0.001$ for all). All scores on subjective health questionnaires (e.g. PHQ-9, HAF-17, WHO-5, SF-12 and IRES-24) were significantly improved with SES to a mild to moderate extent. The changes in the physical and mental scales of the SF-12 (+5.7/+6.1 points) and IRES-24 (physical health +1.2, mental health +1.4, pain 1.0) and in the WHO-5 (+18.3%-points) exceeded the MID. However, the proportion of patients who rated their occupational prognosis negatively increased significantly from 41% to 45% (Table 2).

### Follow-up data

In the follow-up survey, 864 patients (68.5%) returned to work and 67 patients (5.3%) reported having retired, while 79 (6.3%) had applied for pension. Eighty nine patients (7.1%) were unemployed six months after CR and 190 (15.1%) were still on sick leave. Furthermore, 194 (15.4%) patients had experienced readmission to acute hospital care for any cardiac event.

Data for HRQL were available for 1,181 patients (94%). The mean score was 45.5±10 points on the physical component summary scale and 48.9±11 points on the mental component summary scale. Scores for HRQL 6 months after CR were significantly correlated to return to work (age and sex adjusted OR = 1.08/1.06 per point on the physical/mental component summary scale; $p < 0.001$). In general, patients reported a moderately or substantially improved health status due to CR in 935 (74.7%) cases and an enhanced occupational or physical capacity in 876 (69.9%) cases.

### Predictors of occupational resumption and quality of life after cardiac rehabilitation

Acute coronary syndrome as an indication diagnosis and comorbid heart failure upon CR admission were negatively associated with return to work, while a higher endurance training load, HRQL, a high educational level and work stress detected at CR admission had a positive impact on return to work. Patients' pension desire and self-assessed negative occupational prognoses at discharge from CR reduced the probability of returning to work six months after discharge from CR by 67% and 66%, respectively. Additionally, an improvement in mental HRQL by 10 points between CR admission and discharge resulted in an improved probability of returning to work by 30%, whereas an enhancement of heart-focussed anxiety during CR reduced the chances of returning to work by 29% per point difference on the HAF-17. However, the latter association achieved no statistical significance (Fig 2).

Patients' HRQL after six months was primarily predicted by patient-reported outcome measures, which were mostly assessed at admission to CR. In addition, modifications during CR in endurance training load, heart-focussed anxiety, physical quality of life and health were predictive of physical component summary scores, while changes in anxiety, well-being, mental health and general self-efficacy expectations were significant for the mental component summary score at follow-up. For example, an improvement in mental health (IRES-24) of two points during CR resulted in an enhanced MCS of 2.54 points at follow-up (Fig 3).

**Table 1. Baseline characteristics (*n* = 1,262).**

| Characteristics (1) | mean±SD or *n* (%) |
|---|---|
| *Sociodemographic data* | |
| Age (years) | 54.2±7.0 |
| Sex (male) | 968 (76.7) |
| Education *n = 1,249* | |
| <10$^{th}$ grade | 209 (16.8) |
| Secondary school | 691 (55.3) |
| College/university | 288 (23.1) |
| Living situation *n = 1,248* | |
| Family/partner | 987 (79.1) |
| Living alone | 218 (17.5) |
| Other | 43 (3.4) |
| Occupation (employed) *n = 1,248* | 1127 (90.3) |
| Sick leave before CR *n = 1,248* | 892 (71.5) |
| *Cardiac rehabilitation* | |
| Admission to CR | |
| After an acute event | 1044 (82.7) |
| For a chronic disorder | 218 (17.3) |
| Setting of CR | |
| Inpatient | 1158 (91.8) |
| Outpatient/day care | 104 (8.2) |
| *Indication for referral to CR* | |
| ACS | 500 (39.6) |
| Stable CAD | 229 (18.1) |
| Heart valve disease | 153 (12.1) |
| CABG | 96 (7.6) |
| Venous disease | 49 (3.9) |
| Cardiac arrhythmia | 49 (3.9) |
| Aortic diseases | 47 (3.7) |
| Arterial hypertension | 46 (3.6) |
| Chronic heart failure | 36 (2.9) |
| Atherosclerosis (incl. PAD) | 24 (1.9) |
| Intervention (PCI, ICD/CRT) | 19 (1.5) |
| Myo-/endo-/pericarditis | 7 (0.6) |
| Other | 7 (0.6) |
| *Comorbidities/ risk factors* | 3.4±1.6* |
| Arterial hypertension | 840 (66.6) |
| Hyperlipidaemia | 772 (61.2) |
| Diabetes mellitus | 206 (16.3) |
| Atrial fibrillation | 111 (8.8) |
| PAD | 64 (5.1) |
| Depression | 55 (4.4) |
| COPD | 52 (4.1) |
| Kidney disease | 51 (4.0) |

*number of comorbidities; ACS, acute coronary syndrome; CABG, coronary artery bypass graft; CAD, coronary artery disease; COPD, chronic obstructive pulmonary disease; CR, cardiac rehabilitation; CRT, cardiac resynchronisation therapy; ICD, implantable cardioverter defibrillator; PAD, peripheral artery disease; PCI, percutaneous coronary intervention; SD, standard deviation

**Table 2. Functional parameters and risk factors and patient-reported outcome measures at baseline and discharge from cardiac rehabilitation (N = 1,262; mean ± standard deviation; n and percentage).**

| Parameter | CR admission | CR discharge | SES | P-value |
|---|---|---|---|---|
| Cardiovascular risk factors | | | | |
| Smoking behaviour[‡] (smoker) n = 1,234 | 427 (34.6) | 187 (15.2) | - - | <0.001 |
| Systolic blood pressure (mmHg) n = 1,258 | 128.8±18.5 | 121.6±13.8 | 0.39 | <0.001 |
| Diastolic blood pressure (mmHg) n = 1,258 | 80.3±11.4 | 75.2±9.1 | 0.44 | <0.001 |
| LDL-Cholesterol (mmol/l) n = 981 | 4.6±2.5 | 3.8±2.2 | 0.32 | <0.001 |
| Physical performance | | | | |
| Maximum exercise capacity (Watt) n = 790 | 111.4±37.6 | 130.9±41.5 | 0.52 | <0.001 |
| Endurance training load (Watt) n = 1,204 | 48.6±20.6 | 69.7±26.3 | 1.03 | <0.001 |
| 6-min walking distance (m) n = 812 | 451.8±90.6 | 526.6±90.8 | 0.82 | <0.001 |
| Patient-reported outcome measures | | | | |
| Depression (PHQ-9) n = 1,149 | 6.4±4.8 | 4.4±4.0 | 0.43 | <0.001 |
| Heart-focussed anxiety (HAF-17) n = 1,102 | 1.5±0.6 | 1.3±0.6 | 0.32 | <0.001 |
| HAF-17 heart-focussed fear n = 1,146 | 1.6±0.7 | 1.4±0.7 | 0.27 | <0.001 |
| HAF-17 heart-focussed avoidance n = 1,163 | 1.5±1.0 | 1.0±0.8 | 0.44 | <0.001 |
| HAF-17 heart-focussed attention n = 1,158 | 1.3±0.7 | 1.3±0.6 | 0.02 | 0.308 |
| Quality of life/subjective well-being | | | | |
| WHO-5 n = 1,180 | 51.2±25.6 | 69.5±21.1 | 0.71 | <0.001 |
| SF-12 PCS n = 1,072 | 38.9±10.6 | 44.6±9.5 | 0.53 | <0.001 |
| SF-12 MCS n = 1,072 | 48.2±11.9 | 54.3±8.9 | 0.51 | <0.001 |
| IRES-24 physical health n = 1,173 | 5.8±2.7 | 7.0±2.4 | 0.43 | <0.001 |
| IRES-24 mental health n = 1,190 | 6.4±2.5 | 7.8±2.1 | 0.58 | <0.001 |
| IRES-24 pain n = 1,190 | 6.3±2.6 | 7.3±2.4 | 0.40 | <0.001 |
| General self-efficacy expectations (ASKU) | 4.1±0.7 | 4.1±0.7 | 0.10 | <0.001 |
| Lifestyle change motivation (certain/fairly certain) n = 1,187 | 939 (79.1) | 1037 (87.4) | - - | <0.001 |
| Self-assessed health prognosis (excellent/very good) n = 1,187 | 509 (42.9) | 605 (51.0) | - - | <0.001 |
| Pension desire (yes) n = 1,170 | 205 (17.5) | 182 (15.6) | - - | 0.028 |
| Self-assessed occupational prognosis (negative) n = 1,137 | 463 (40.7) | 507 (44.6) | - - | <0.001 |

[‡]Patients who quit smoking due to the acute event before subsequent cardiac rehabilitation were classified as smokers upon admission to rehabilitation.

ASKU, Allgemeine Selbstwirksamkeit Kurzskala (short scale for measuring general self-efficacy beliefs); CR, cardiac rehabilitation; HAF-17, Herzangstfragebogen (German version of the Cardiac Anxiety Questionnaire); IRES-24, indicators of rehabilitation status; LDL, low density lipoprotein; PHQ-9, Patient Health Questionnaire; SES, standardised effect size according Cohen's d; SF-12, Short-Form health survey with PCS, physical component summary and MCS, mental component summary; WHO-5, World Health Organization well-being index

## Discussion

This investigation found that 69% of patients successfully returned to work 6 months after CR. HRQL was significantly improved by seven percent (physical component) and one percent (mental component) compared to CR admission. The OutCaRe study demonstrated that occupational reintegration, as well as HRQL are predominantly determined by patient-reported outcome measures, more than by clinical parameters, cardiovascular risk factors or physical performance. Specifically, pension desire, negative patient expectations of occupational resumption, acute coronary syndrome, comorbidity (i.e. heart failure) and heart-focussed anxiety represented barriers to occupational reintegration, while an enhanced endurance training load and quality of life on the SF-12, high educational level and work stress facilitated return to work. HRQL in its physical dimension was primarily influenced by physical health and quality of life, heart-focussed anxiety at CR admission and changes during CR, as well as self-assessed

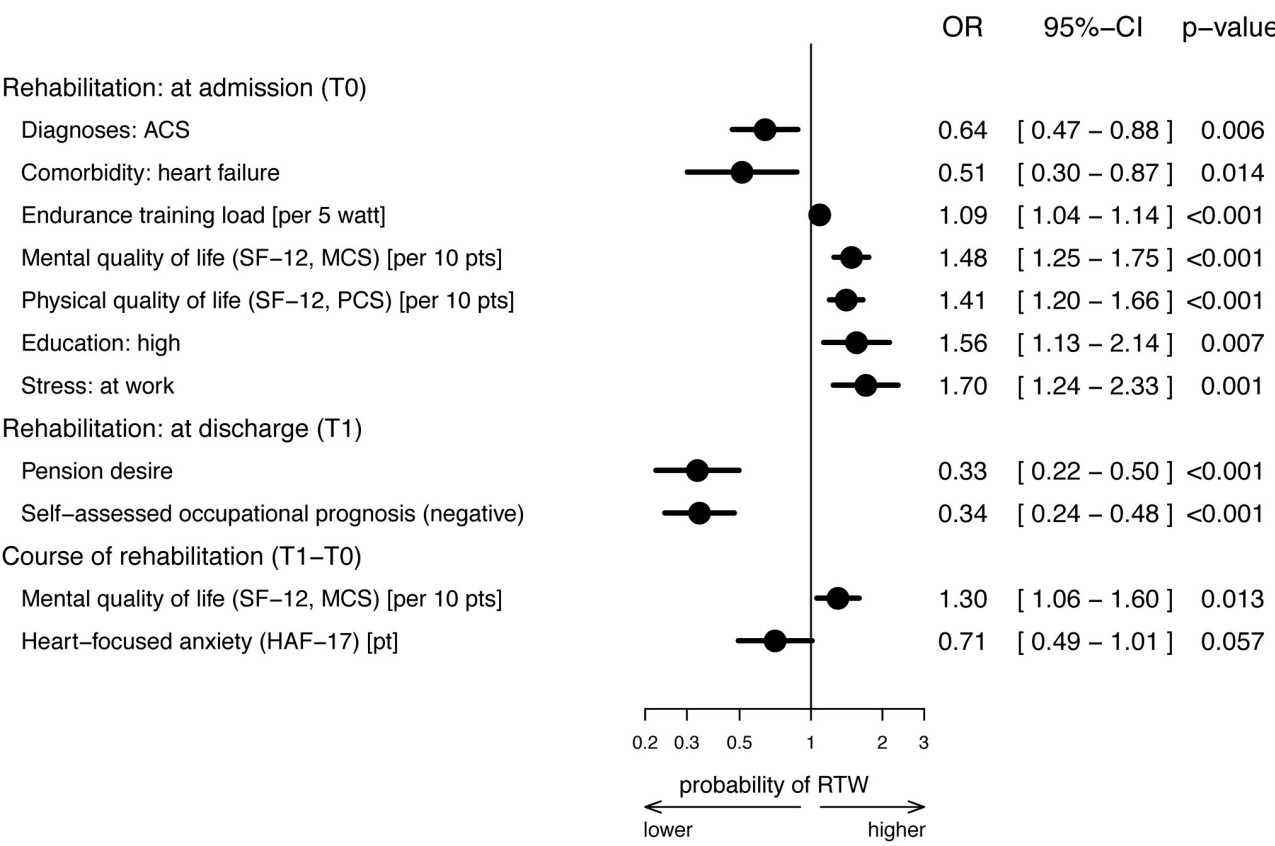

**Fig 2. Predictors of returning to work after cardiac rehabilitation, imputed model.** The forest plot shows the final model after backward selection. The following variables were taken into account in the starting model (see S3 Fig): at admission to rehabilitation: sex, smoking, lifestyle change motivation, pension desire, self-assessed occupational prognosis, self-assessed health prognosis, living situation, educational level, rehabilitation referral for chronic disorder, comorbidities (diabetes mellitus, depression, peripheral artery disease, heart failure, diseases of the back and spine, stress at work, stress from major life events, age, systolic/diastolic blood pressure, body mass index, endurance training load, depression (PHQ-9), heart-focussed anxiety (HAF-17), well-being (WHO-5), physical/mental component summary on the SF-12, physical/mental health and pain in the IRES-24, self-efficacy (ASKU); at discharge from rehabilitation: smoking, lifestyle change motivation, pension desire, self-assessed occupational prognosis, self-assessed health prognosis; changes during rehabilitation: systolic/diastolic blood pressure, body mass index, endurance training load, depression (PHQ-9), heart-focussed anxiety (HAF-17), well-being (WHO-5), physical/mental component summary on the SF-12, physical/mental health and pain on the IRES-24, self-efficacy (ASKU). T0 –baseline measurement at CR admission, T1 –CR discharge, T2 –follow-up six months after CR discharge. CI, confidence interval; HAF-17, Herzangstfragebogen (German version of the Cardiac Anxiety Questionnaire); OR, odds ratio; pt(s), point(s); RTW, return to work; SF-12, Short-Form health survey with PCS, physical component summary and MCS, mental component summary.

health prognosis and pension desire at CR discharge. Psychoemotional parameters, for instance the mental health scale of the IRES-24, perceived stress, comorbid depression, general self-efficacy and well-being assessed by the WHO-5, predicted the mental component of HRQL on the SF-12 as might be expected. Heart-focussed anxiety, pension desire and mental and physical HRQL at different measurement times in CR were the only predicting parameters for both return to work and HRQL after CR.

Heart-focussed anxiety as measured by the HAF-17 comprises the specific fear of recurrent cardiac events, heightened attention to cardiac-related stimuli and symptoms, as well as the aspect of avoidance. [27] These conditions may lead to a reduced adherence to exercise and medication intake and consequently to increased cardiac events.[40] Van Beek et al. found the prognostic impact particularly driven by avoidance behaviour and identified the special need to address cardiac anxiety in CR programs.[40] In fact, the sum score of heart-focussed anxiety and especially the avoidance subscale showed significant improvements to a small extent in

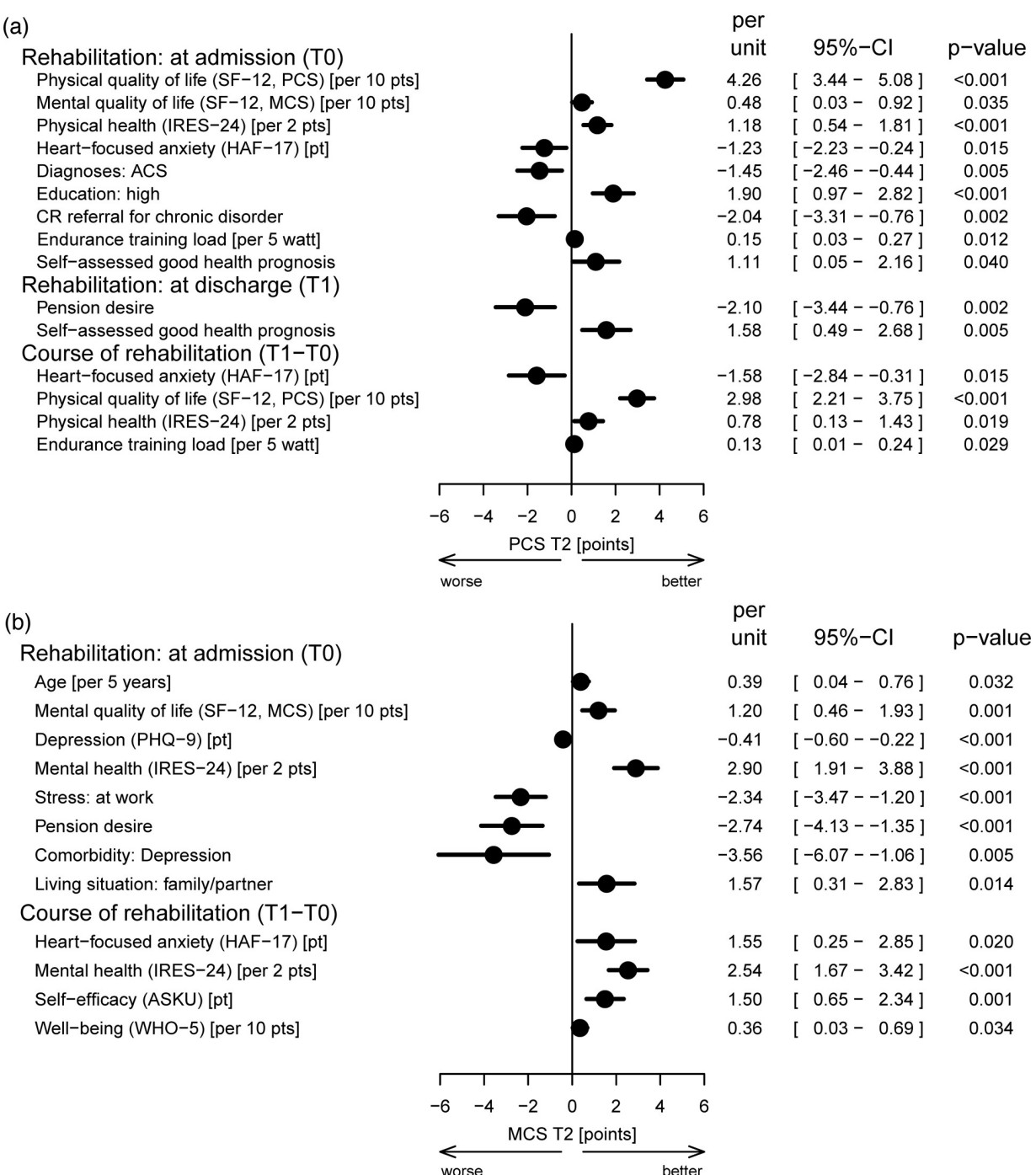

**Fig 3. Predictors of health-related quality of life after cardiac rehabilitation, imputed model.** The forest plots show the final models after backward selection. The following variables were taken into account in the starting models (see S1 and S2 Figs): at admission to rehabilitation: sex, smoking, lifestyle change motivation, pension desire, self-assessed occupational prognosis, self-assessed health prognosis, living situation, educational level, rehabilitation referral for chronic disorder, comorbidities (diabetes mellitus, depression, peripheral artery disease, heart failure, diseases of the back and spine, stress at work, stress from major life events, age, systolic/diastolic blood pressure, body mass index, endurance training load, depression (PHQ-9), heart-focussed anxiety (HAF-17), well-being (WHO-5), physical/mental component summary on the SF-12, physical/mental health and pain on the IRES-24, self-efficacy (ASKU); at discharge from rehabilitation: smoking, lifestyle change motivation, pension desire, self-assessed occupational prognosis, self-assessed health prognosis; changes during rehabilitation: systolic/diastolic blood pressure, body mass index, endurance training load, depression (PHQ-9), heart-focussed anxiety (HAF-17), well-being (WHO-5), physical/mental component summary on the SF-12, physical/mental health and pain on the IRES-24, self-efficacy (ASKU). T0 –baseline measurement at CR admission, T1 –CR discharge, T2 –follow-up 6 months after CR discharge. ACS, acute coronary syndrome; ASKU, Allgemeine

Selbstwirksamkeit Kurzskala (short scale for measuring general self-efficacy beliefs); CI, confidence interval; HAF-17, Herzangstfragebogen (German version of the Cardiac Anxiety Questionnaire); IRES-24, indicators of rehabilitation status; OR, odds ratio; PHQ-9, Patient Health Questionnaire; pt(s), point(s); RTW, return to work; SF-12, Short-Form health survey with PCS, physical component summary and MCS, mental component summary; WHO-5, World Health Organization well-being index.

the investigated population during the three- to four-week CR programme, which is in line with other recent studies.[41,42]

'Pension desire' was assessed in our study by a single question on the intention to apply for retirement without any causal differentiation. Thus, this parameter may implicate aspects of non-captured occupation-related factors, such as non-motivation to return to work, occupational self-efficacy, job satisfaction, job control or expectations of retirement, as they are identified to predict occupational resumption in other publications.[13,43] However, patients with pension desire and negative self-assessed occupational prognoses in CR suffer from a high burden of predominantly social and mental disease consequences. On the other hand, applied coping strategies and support services are mainly focussed on physical aspects of the cardiovascular disease. [44] Especially the essential adaptation of everyday life including the work environment after a cardiac event and subsequent CR could be overtaxing, triggering the desire for early retirement.

In this context, social workers in CR play a crucial role as they advise patients, for example on retraining or gradual reintegration as needed. The support by social workers in CR programmes is mandatory according to the specifications of the German pension insurance. Nevertheless, in accordance with previous studies and official data, return to work rates after a cardiac event were only 69% in our study. [8,13,45] An original randomised clinical trial evaluating an extended social therapy counseling and training program during CR still had no effect on return to work and HRQL in patients at risk of occupational reintegration. [16] This finding suggests that the regular therapy density in the standardised German intensive short-term CR program as described in the methods section is at maximum volume. Continuous support by a social worker in the further course after a cardiac event could remedy this.

The impact of pension desire on HRQL on both the physical and mental scales after CR in our study is less intuitive. It can be interpreted in the context of patients' expected (likely occurring) difficulties in everyday life and working environment, whereas a mediating role of pension desire for return to work and HRQL is conceivable. Van Cauter et al. accordingly evidenced the significant association between return to work and an enhanced HRQL in the EUROASPIRE IV study. [15]

Also, the counterintuitive effects of perceived stress on occupational reintegration are remarkable. Thus, perceived stress at work facilitated patients' occupational reintegration. Our findings go along with an Iranian study of 248 patients after coronary bypass surgery, where the probability of returning early to work was 2.3 times higher in patients with job stress. [46] We assume that patients who report a higher perception of work stress may hold a leading position with a higher degree of personal identification and recognition, resulting in a greater motivation to return to work. This interpretation is quite speculative, but may be supported by the fact that the proportion of patients who returned to work was significantly higher by 14 percentage points in highly educated versus less-educated patients (78% vs. 64%). However, perceived stress at work was not associated with the educational level and, in addition, predicted a diminished mental quality of life in the present investigation.

Furthermore, attention should be paid to changes during CR in the parameters independently affecting the analysed mid-term outcome of CR. Besides heart-focused anxiety as mentioned above, the means that mental and physical quality of life as well as subjective health

(IRES-24) and well-being on the WHO-5 were significantly improved, with small to moderate effect sizes. The notified changes are considered clinically relevant since they substantially exceed the MID by 1.8 (WHO-5) to 2.9 times (physical component of SF-12).

For the modification of this multiplicity of influencing parameters, in particular patient-reported outcome measures, CR should be composed as a multi-modal programme based on the holistic approach of the biopsychosocial model of the International Classification of Functioning, Disability and Health by the World Health Organization (WHO) [47] and in accordance with current national and international guidelines. [1,48,49] In particular, close cooperation in an interprofessional CR team including cardiologist, social worker and sports therapist or physiotherapist is needed. The offered CR programme should be individualised in the sense of patient-shared decision making to achieve the best possible outcomes.

## Limitations

This work is limited in terms of various methodological aspects. First, our recruitment quota was 32%, which can reduce the generalisability of our results to the general CR population. However, our participants were similar to our target population (patients in CR aged up to 65 years) according to statistics on the rehabilitation services of the German pension insurance with regard to age and sex. [50] Second, there is a selection bias due to the incomplete follow up. Non-responding study participants were younger, less educated, more likely to smoke and less frequently employed. At discharge from CR, they reported a lower health status. These differences were taken into account by the imputation model. However, there may be reasons for the absence that were not covered by our data. This has to be kept in mind when interpreting the study results. Furthermore, we investigated a heterogeneous population including patients of all diagnoses in CR. The main objective of the OutCaRe project was to identify and evaluate performance measures for the entire spectrum of CR as generally offered in Germany. Therefore, specific statements pertaining to individual indication groups cannot be derived from the current study. In addition, the feasibility of a parameter as a performance measure was assessed in terms of data availability. [21] Patients were allowed to refrain from answering parts of the questionnaires or individual questions. Missing values were imputed in the statistical models to balance the data. However, it may be that these modeling approaches do not compensate for the selection effects. Finally, the SF-12 used to operationalise HRQL is a generic instrument. Probably, a disease-specific screening tool (eg. HeartQoL questionnaire) [51] could achieve a higher responsiveness to change and acceptance in the investigated heart-disease population. Nevertheless, the SF-12 is well established and commonly used, which ensures the comparability of our results with other studies and populations.

Multicollinearity may limit the interpretability of model coefficients. However, the variance inflation factors of the studied predictors were distinctly below the generally accepted limit of 10 with one exception: mental health at admission had a variance inflation factor of 10.48. Altogether, the multivariate models were sufficiently stable to allow clinical interpretations.

## Conclusion

Return to work and physical, as well as mental HRQL half a year after CR were predominantly determined by patient-reported outcome measures, whereas patients' pension desire and heart-focussed anxiety had a dominant impact on all investigated endpoints. Changes in patient-reported outcome measures during CR affected the occupational and health-related prognosis, underscoring the importance of the multi-component approach in CR. Therefore, questionnaires assessing subjective health should be applied consequently upon commencing

rehabilitation with findings subsequently affecting the individual therapy content and rehabilitation goals.

## Supporting information

**S1 Fig. Full model-multiple imputation(n = 1181).** SF—12, MCS—T2.
(TIF)

**S2 Fig. Full model-multiple imputation(n = 1181).** SF—12, PCS—T2.
(TIF)

**S3 Fig. Full model-multiple imputation(n = 1262).** Return to work(RTW).
(TIF)

**S4 Fig. Full model-multiple imputation(n = 1262).** No significant different probabilities for subdiagnoses (p = 0.127).
(TIF)

## Acknowledgments

We thank all participating patients, the frontline staff of the cooperating CR centres and the OutCaRe investigators: Heinz Völler, Klinik am See, Rüdersdorf (principal investigator; e-mail: heinz.voeller@klinikamsee.com); Johannes Glatz and Eike Langheim, Reha-Zentrum Seehof, Teltow; Axel Schlitt, Paracelsus-Harz-Klinik, Quedlinburg; Jörg Nothroff, MediClin Reha-Zentrum Spreewald, Burg; Klaus Schröder, ZAR Stuttgart, Stuttgart; Ronja Westphal, Segeberger Kliniken, Bad Segeberg; Christa Bongarth, Klinik Höhenried, Bernried; Sieglinde Spörl-Dönch and Gerhard Alexander Müller, Frankenklinik, Bad Neustadt; Markus Wrenger, Caspar Heinrich Klinik Bad Driburg GmbH & Co. KG, Bad Driburg; Roger Marx, MediClin-Fachklinik Rhein/Ruhr, Essen; Rainer Schubmann, Dr. Becker Klinik Möhnesee, Möhnesee; Martin Schikora, Brandenburgklinik, Bernau; all centres are located in Germany. We gratefully acknowledge the German Society of Prevention and Rehabilitation of Cardiovascular Diseases (DGPR) for their cooperation. We thank Klaus Balzer for his help in creating the plots and tables.

## Author Contributions

**Conceptualization:** Annett Salzwedel, Axel Schlitt, Christa Bongarth, Rona Reibis, Heinz Völler.

**Data curation:** Annett Salzwedel, Iryna Koran.

**Formal analysis:** Susanne Sehner, Karl Wegscheider.

**Funding acquisition:** Annett Salzwedel, Heinz Völler.

**Investigation:** Iryna Koran, Eike Langheim, Axel Schlitt, Jörg Nothroff, Christa Bongarth, Markus Wrenger, Heinz Völler.

**Methodology:** Annett Salzwedel, Susanne Sehner, Karl Wegscheider.

**Project administration:** Annett Salzwedel, Heinz Völler.

**Writing – original draft:** Annett Salzwedel, Susanne Sehner, Rona Reibis, Karl Wegscheider.

**Writing – review & editing:** Annett Salzwedel, Rona Reibis, Karl Wegscheider, Heinz Völler.

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
