## [Decision Letter · Decision Letter 0]

21 Nov 2019

PONE-D-19-29345

Patient-reported outcomes predict return to work and health-related quality of life 6 months after cardiac rehabilitation: Results from a German multi-centre registry (OutCaRe)

PLOS ONE

Dear Dr. Salzwedel,

Thank you for submitting your manuscript to PLOS ONE. After careful consideration, we feel that it has merit but does not fully meet PLOS ONE’s publication criteria as it currently stands. Therefore, we invite you to submit a revised version of the manuscript that addresses the points raised during the review process.

Return to work (RTW) and quality of life following a cardiac event are important patient outcomes and this paper looks to identify factors that predict these outcomes following cardiac rehabilitation. The paper has been assessed by two experts in the field of cardiac rehabilitation and they have raised some excellent points. While the paper identifies an important area of research, the paper requires major revisions before it can be considered eligible for publication. Below you will find comments from the reviewers, as well as some points of my own. The authors should also consider all of the work done on predictors of HRQL in the CR setting and describe how this work adds to the pre-existing literature. If you feel that you can make the required changes, we would be happy to re-review the paper.

We would appreciate receiving your revised manuscript by Jan 05 2020 11:59PM. To enhance the reproducibility of your results, we recommend that if applicable you deposit your laboratory protocols in protocols.io, where a protocol can be assigned its own identifier (DOI) such that it can be cited independently in the future. For instructions see: http://journals.plos.org/plosone/s/submission-guidelines#loc-laboratory-protocols

We look forward to receiving your revised manuscript.

Kind regards,

Stephanie Prince Ware, PhD

Academic Editor

PLOS ONE

Journal Requirements:

3. One of the noted authors is a group or consortium: OutCaRe investigators

In addition to naming the author group, please list the individual authors and affiliations within this group in the acknowledgments section of your manuscript. Please also indicate clearly a lead author for this group along with a contact email address.

Additional Editor Comments:

1. Abstract: The methods should state more clearly that you are looking to predict RTW with the variables collected at baseline and discharge from cardiac rehabilitation. Also, the results should present data.

2. Line 64: what is the advantage?

3. Please use acronyms consistently. For example RTW is used on line 64, but spelled out fully on 67. HRQL is spelled out in full on line 80. Lines 95-96 have both of these terms spelled out fully.

4. Methods: Consider using headings for all of the parameters and describing them under each heading to help improve readability. It is difficult to follow all of the measures.

5. Reason for CR should also be controlled for in the model as the severity of event is likely to impact an individual’s desire to RTW.

6. It sounds like the CR program is only 3-4 weeks in duration which appears quite short. The authors describe 12 training sessions per week, but this does not appear to be feasible. Can you clarify?

7. What activities are the participants undertaking when they are asked for their BORG rating?

8. The SF-12 is a generic measure of HRQL for the outcome following CR. This is likely a limitation as it does not assess disease-specific HRQL.

9. Is there an ethics approval number associated with this project?

10. Were the RTW analysis only conducted on those who were working at baseline?

11. Statistics: Please change “metric” with “descriptive”.

12. Line 152: Respectively to what?

13. Lines 197-198: In the methods clinically important differences in these measures should be outlined and results should be discussed in terms of clinical and statistical significance.

14. Why are the ORs for RTW age-adjusted? There is a limitation of age on the participants already.

15. Include all outcomes in figures 2 and 3.

16. Line 285, what does “opp. RTW motivation” mean?

17. Discussion: There is discussion that greater reported stress at work may be related to earlier RTW due to those individuals holding higher positions. The study collected respondent education; this should be looked at in relation to stress and RTW.

18. Line 311: this appears to be an error: “physis-focussed”

19. It is intuitive that physical health measures would predict the PCS of the SF-12 while mental health measures would predict the MCS of the SF-12. The authors should comment on which aspects are the most important. Also, there is likely multicollinearity occurring in the models, was this assessed?

20. Discussion: the authors should consider discussing the benefit of Social Workers as part of the interdisciplinary treatment in CR.

21. Discussion: Were the data missing at random? Were those with complete data different than those with incomplete data?

22. Reference 1 and 35 are the same.

Reviewers' comments:

Reviewer's Responses to Questions

**Comments to the Author**

1. Is the manuscript technically sound, and do the data support the conclusions?

Reviewer #1: Yes

Reviewer #2: Partly

2. Has the statistical analysis been performed appropriately and rigorously? 

Reviewer #1: I Don't Know

Reviewer #2: I Don't Know

3. Have the authors made all data underlying the findings in their manuscript fully available?

Reviewer #1: No

Reviewer #2: No

4. Is the manuscript presented in an intelligible fashion and written in standard English?

Reviewer #1: Yes

Reviewer #2: Yes

5. Review Comments to the Author

Reviewer #1: This paper has hallmarks of good clinical and ecological validity in light of its links with a patient outcome registry. It would be helpful if the Abstract results included actual data, as opposed to statements which are more like conclusions

Whilst desire to retire was an interesting and important covariate to RTW, would not age also be key, in respecting the Methods clearly had a cut-off of 65 years.

There are a number of risk factor data reported and mention of the Borg scale. However, their relevance to all this isn't clear. Furthermore, the changes in most risk factors, were these actually related to the Rehab process or merely functions of medications prescribed.

Line 105 mentions attendance of 12 training units per week? What is a training unit? In terms of programme fidelity and in addition to programme uptake and completion, how compliant were the participants with attending? (e.g. participation rates per week). The Exercise Capacity was measured but not factored into the relationship with RTW? A way to check on whether exercise capacity had changed was to see if patients heart rate or RPE was lower for any give workmate in Watts (submit or max), which in fact you have reported

More clarity required whether RTW data included those already unemployed and or on sick leave prior to commencement of CR (apologies for my oversight if this has been stated)

There seem to be lots of abbreviations; can these be reduced? or at least make sure they are clearly defined in legends for all Tables and Figures

Reviewer #2: This paper aimed to identify predictors of return to work (RTW) and quality of life (QOL) 6 months after cardiac rehabilitation. Although the paper describes an interesting topic, I do have quit some concerns and comments that need attention. Some major revisions need to be made in the manuscript before this manuscript can be reviewed in detail.

Major comments:

1. The authors should be more clear what is the novelty of their paper. In the introduction the authors describe that several factors are known to influence RTW (such as depression, anxiety, expectations) in lines 69-72. However, in line 77 it is said that few is known about the association between patient centered outcomes and RTW. Are the factors mentioned in lines 69-72 not also patient centered outcomes? And in line 62, it is described that the effect of CR on QOL is well-known, so why is QOL choses as an outcome in this paper?

2. The methods section describing the study outcomes/ predictors is extremely difficult to follow (lines 116-136), partly due to the large amount of parameters. Could the authors try to re-organize this paragraph and make clear what exactly is measured and which instrument is used (for instance with a table or using bullet points).

3. Could the authors give some more information on the content of the CR program (duration, training sessions and counselling sessions). Twelve training sessions per week seem quite high.

4. With regard to the used statistics I have some questions:

a. Were all predictors added to the same multiple linear regression model? I can imagine that there might be some overlap and correlation between some of the predictors. Did the authors test for multicollinearity?

b. Could the authors add some information about the power of their study to test for this large amount of predictors?

c. Why was chosen for only one imputation and not for multiple imputations?

5. With regard to generalizability of the data: could the authors say something about patients that were not willing to participate in this study? Is the sample that was willing to participate representative of the whole population?

6. Please clarify in your paper what is seen as a PROM and what is seen as subjective health (and what is the difference between these two concepts).

7. In the paper there is a lot of attention focused on the role of pension desire in RTW. What is the rationale behind investigating this? If patients have the desire and possibility to retire, should that be seen as a negative outcome of CR? In my opinion, it should not be seen like that. It can be a good and well thought out choice. Reading your results, one could conclude that to improve RTW we should try to change someone’s wish to retire during CR. What is the vision of the authors on this?

8. The authors chose to look for predictors of successful return to work. On top of patients still on sick leave, the group of patient with unsuccessful return to work included patients that chose to retire, patient that applied for pension and patient that were already unemployed before CR. Wouldn’t it be more interesting to look for predictors of patients that failed to return to work as a result of their cardiac disease, even though they would have wanted to (the group of 15% still on sick leave)? This would be the group that might benefit of extra support during or after CR.

9. The percentages in lines 215-217 add up to 102.3%.

10. Lines 228-251 in the results are difficult to follow. Could the authors clarify if the mentioned predictors are measured at start or completion of CR or concern an improvement during CR?

11. If I understand well, figures 2 and 3 only show outcomes of significant results (predictors). This gives a misleading representation of all results. I would suggest to add all variables that were tested to this figure so that it is clear which predictors were significant and which were not.

12. The discussion is quite difficult to follow. When are results of other authors discussed and when does it concern the author’s own results? Also, it is sometimes difficult to follow if the outcomes with regard to QOL or RTW are discussed. And how can these results be used in daily clinical practice. I suggest a re-organization of this section (first discuss own results (split RTW and QOL), second how does this compare to results of other studies, third what can be done differently with this information in daily CR practice) and a “deeper” discussion of the work.

Smaller remarks:

1. Please try to be consistent in your manuscript with the use of abbreviations.

2. What do the authors mean with “mid-term associations”?

3. Could the authors try to rephrase the aim in the abstract? It is a complex sentence that I had to read several times to understand.

4. What do the authors mean with occupational study? (line 29)

5. In Table 1 :

a. living situation does not add up to 100%.

b. What is meant with the number 3.4 +/- 1.6 behind comorbidities?

c. Is AF a diagnosis or comorbidity?

6. In Table 2: could the author add p-values?

7. Social integration is not the same as return to work (it also includes return to leisure time activities etc). I would suggest to not mix these terms.

8. How many centres were involved in this study?

9. The description of the figures has a lot of overlap with the text.

6. PLOS authors have the option to publish the peer review history of their article (what does this mean?). If published, this will include your full peer review and any attached files.

Reviewer #1: No

Reviewer #2: No

---

## [Author Response · Author response to Decision Letter 0]

17 Jan 2020

Please see the attached response to the reviewers letter, that includes a formatted detailed point-by-point reply.

At first, we thank the editor and reviewers for reading the manuscript as well as for their helpful comments, with which we have dealt intensively. We reviewed and revised the manuscript accordingly. Please find our point-by-point response below. Modifications in the manuscript are highlighted in grey. 

Sincerely,

Annett Salzwedel

for the authors of the manuscript.

Editor

Return to work (RTW) and quality of life following a cardiac event are important patient outcomes and this paper looks to identify factors that predict these outcomes following cardiac rehabilitation. The paper has been assessed by two experts in the field of cardiac rehabilitation and they have raised some excellent points. While the paper identifies an important area of research, the paper requires major revisions before it can be considered eligible for publication. Below you will find comments from the reviewers, as well as some points of my own. The authors should also consider all of the work done on predictors of HRQL in the CR setting and describe how this work adds to the pre-existing literature.

…

*****We tried to analyse all available studies on the same topic. In more than 30 relevant original research publications and additional reviews, which were evaluated in detail, there were no studies, that systematically investigated modifiable parameters in the CR setting. We rephrased and extended the second part of the introduction to describe the purpose of our work.

Additional Editor Comments:

1. Abstract: The methods should state more clearly that you are looking to predict RTW with the variables collected at baseline and discharge from cardiac rehabilitation. Also, the results should present data.

*****We rewrote the abstract including the rephrasing of the study aim. The results include now several data. Due to the word restriction, we present only a selection of the most important findings. 

2. Line 64: what is the advantage?

*****In contrast to exercise-based rehab, comprehensive rehabilitation programs reduce not only cardiovascular mortality and myocardial infarction but also all-cause mortality and stroke rate according to the review and meta-analysis by Halewijn et al. The multi-modal approach also increases return to work up to 6 months (Hegewald et al.). We modified the sentence as follows:

“However, CR programmes based on a multi-modal approach seem to be at an advantage over exercise-based programmes regarding both clinical and occupational outcomes.[4,6]”

3. Please use acronyms consistently. For example RTW is used on line 64, but spelled out fully on 67. HRQL is spelled out in full on line 80. Lines 95-96 have both of these terms spelled out fully.

*****We revised the manuscript regarding the consistent use of abbreviations. Apart from the abstract, we also omit the abbreviation of return-to-work and patient reported outcome measures in the manuscript for better readability.

4. Methods: Consider using headings for all of the parameters and describing them under each heading to help improve readability. It is difficult to follow all of the measures.

*****We re-organized the paragraph using bullet points, as suggested by Reviewer # 2, hopefully this will make it easier to read the manuscript.

5. Reason for CR should also be controlled for in the model as the severity of event is likely to impact an individual’s desire to RTW.

*****The CRF distinguished between 12 different reasons for CR. Some of these reasons were rare. Correspondingly, this variable was not selected for the final RTW model. To control the reason for CR, we simplified the variable to ‚ACS‘ vs ‚no ACS‘. This variable was selected for the final model. Although not significant, we added the marginal means of RTW by reason for CR to the supplement (figure S4).

6. It sounds like the CR program is only 3-4 weeks in duration which appears quite short. The authors describe 12 training sessions per week, but this does not appear to be feasible. Can you clarify?

*****We extended the description of the conducted rehab program as provided by the German pension insurance and we hope for a better understanding: 

“All patients performed a comprehensive standardised CR programme with a regular duration of three to four weeks according to the specifications of the German pension insurance.[16] The programme can be performed in either an inpatient or outpatient setting and consists of all-day activities including counseling by a cardiologist, risk-factor modification strategies (e.g. patient education on nutritional habits, smoking cessation, physical activity, and medication adherence), physician-supervised exercise training and sports therapy (e.g. training on a bicycle ergometer, outdoor walking, resistance training, gymnastics), psychosocial interventions (health education and counseling, psychotherapy, stress management in single or group sessions), and vocational assessment and physician and social worker counseling. [17,18] On average, patients perform 12 training and sports therapy units per week with a duration –depending on the training group and physical performance – up to 30 Minutes and 45 Minutes for outdoor walking, respectively, and 8 additional counselling sessions.[19]”

7. What activities are the participants undertaking when they are asked for their BORG rating?

*****BORG rating was done during the endurance training on the bicycle ergometer. However, the BORG rating is of no relevance for the presented results – we removed this information from the text.

8. The SF-12 is a generic measure of HRQL for the outcome following CR. This is likely a limitation as it does not assess disease-specific HRQL.

*****We added this to the limitations:

“The SF-12 used to operationalize HRQL is a generic instrument. Probably, a disease-specific screening tool (eg. HeartQoL questionnaire) [40] could achieve a higher responsiveness to change and acceptance in the investigated heart-disease population. Nevertheless, the SF-12 is well established and commonly used, which ensures the comparability of our results with other studies and populations.”

 

9. Is there an ethics approval number associated with this project?

*****The approval number (S 4(a)/2017) is added to the Ethics section.

10. Were the RTW analysis only conducted on those who were working at baseline?

*****The RTW analysis was conducted to all patients as the occupational resumption is the main goal of CR for the German pension insurance, which is the funding institution for the most CR patients before the retirement age of 65. Therefore, the support by social workers in CR is mandatory according to the specifications of the German pension insurance. Social workers advise for example on retraining or gradual reintegration as needed. We modified the regarding sentence in the statistics: “For all further analyses, only patients with a non-missing 6 month outcome in the original dataset were included, regardless of the employment status at baseline.” (line 201-203) 

11. Statistics: Please change “metric” with “descriptive”.

*****We have changed the corresponding sentence to ‚For description, continuous variables are presented as means ± standard deviation, and categorical variables as absolute values and percentages.‘ to emphasize the different presentation of continuous and categorical variables.

12. Line 152: Respectively to what?

*****We modified the sentence as follows: “Differences in variables between admission to CR (reference), discharge from CR, and follow-up were tested for statistical significance using Wilcoxon for continuous variables and McNemar tests for categorical variables, respectively.” 

13. Lines 197-198: In the methods clinically important differences in these measures should be outlined and results should be discussed in terms of clinical and statistical significance.

*****Minimal (clinically) important difference (MID or MCID) is a concept depending value, mostly calculated to evaluate the longitudinal change of a measure in an individual based on the need for intervention. There are several methods to calculate a MID without a consense for appropriate use within the scientific community. We added available and appropriate MIDs to the statistic section, results and discussion as follows:

“In addition, the changes in patient-reported outcome measures during CR were assessed using the minimal important difference (MID). This concerns to the IRES-24 (MID 0.5 points) [30] and the SF-12 (MID 2 points for the physical and 3 points for the mental component summary, respectively).[31] For the WHO-5, we anticipate a MID of 10%-points.[23] No MID was considered for PHQ-9 because the MID reported in the literature focuses the change of PHQ-values in the acute phase of depression treatment in affected patients,[32] which is of subordinate relevance in our investigation. There is no MID available for the HAF-17.”

“The changes in the physical and mental scales of SF-12 (+5.7/+6.1 points) and IRES-24 (physical health +1.2, mental health +1.4, pain 1.0) and in the WHO-5 (+18.3%-points) exceeded the MID.”

“Furthermore, attention should be paid to changes during CR in the parameters affecting the analysed mid-term outcome of CR. Besides heart-focused anxiety as mentioned above, the means in mental and physical quality of life as well as subjective health (IRES-24) and well-being in the WHO-5 were significantly improved with small to moderate effect sizes. The notified changes are considered clinically relevant since they substantially exceed the MID by 1.8 (WHO-5) to 2.9 times (physical component of SF-12).”

However, as stated in the methods section (line 107-110) this paper focuses on the predictive value of parameters for return-to-work and HRQL. The modifiability of the parameters during CR is expressly subject of previous publication and is comprehensively discussed there. (Zoch-Lesniak B. et al. Performance Measures for Short-Term Cardiac Rehabilitation in Patients at Working Age: Results of the Prospective Observational Multicenter Registry OutCaRe. Archives of Rehabilitation Research and Clinical Translation. 2019. In press.).

14. Why are the ORs for RTW age-adjusted? There is a limitation of age on the participants already.

*****That is right, patients at pension age (>65 years) were excluded from the study. Nevertheless, according to experience, age and sex have an impact on RTW as younger patients have a higher opportunity returning to work. Therefore, we calculated the bivariate association between HRQL 6 months after CR and RTW adjusted for age and sex.

Age was also taken into account conducting the multivariable predictor analysis of RTW, but achieved no statistical significance and was eliminated from the model. 

15. Include all outcomes in figures 2 and 3.

*****We now clarify in the statistics section that we present only final models after backward selection. These models give more precise estimates of regression coefficients than the models that include all potential regressors which are overfitted to the data. However, we show the analysis of all measurements in the supplement (figures S1-S3). 

 

16. Line 285, what does “opp. RTW motivation” mean?

*****It meant ‘the opposite of motivation to return to work’. We removed this misleading phrase.

17. Discussion: There is discussion that greater reported stress at work may be related to earlier RTW due to those individuals holding higher positions. The study collected respondent education; this should be looked at in relation to stress and RTW.

*****High educated patients returned to work in 78%, while the proportion in less educated patients was 64%. There was no association between stress and education. We added the information to the responding paragraph in discussion (line 366-372):

“We assume that patients who report a higher perception of work stress may hold a leading position with a higher degree of personal identification and recognition, resulting in a higher intention to return-to-work. This interpretation is quite speculatively but may be supported by the fact that the proportion of patients returned to work was significantly higher by 14%-points in high educated than in less educated patients (78% vs. 64%). However, perceived stress at work was not associated with the educational level and also predicted a diminished mental quality of life in our study.”

18. Line 311: this appears to be an error: “physis-focussed”

*****We removed the term.

19. It is intuitive that physical health measures would predict the PCS of the SF-12 while mental health measures would predict the MCS of the SF-12. The authors should comment on which aspects are the most important. Also, there is likely multicollinearity occurring in the models, was this assessed?

*****We restructured the entire discussion section since there were items suggested to add. Hereby, we have also realigned the focus of HRQL related predictors.

We studied potential multicollinarity by calculation of the variance inflation factors (VIF) for the complete set of potential regressors. VIF were distinctly below the generally accepted limit of 10 with one exception: mental health at baseline as assessed by use of the IRES-24 had an VIF of 10.48 which means that the corresponding coefficients might be slightly compromised by multicollinearity. We added to the limitations section: “Multicollinearity may limit the interpretability of model coefficients. However, the variance inflation factors of the studied predictors were distinctly below the generally accepted limit of 10 with one exception: the mental health at admission had a variance inflation factor of 10.48. Altogether, the multivariate models were sufficiently stable to allow clinical interpretations.” 

20. Discussion: the authors should consider discussing the benefit of Social Workers as part of the interdisciplinary treatment in CR.

*****We added a discussion of the benefit of social workers in line 345-354: “In this context, social workers in CR play a crucial role as they advise for example on retraining or gradual reintegration as needed. The support by social workers in CR program is mandatory according to the specifications of the German pension insurance. Nevertheless, in accordance with previous studies and official data, return-to-work rates after a cardiac event were only 69% in our study. [7,9,32] An own randomized clinical trial evaluating an extended social therapy counseling and training program during CR remained without an effect on return-to-work and HRQL in patients at risk of occupational reintegration. [12] This finding suggests that the regular therapy density in the standardized German intensive short-term CR program as described in the methods section is of maximum volume. A continuous support by a social worker in the further course after a cardiac event could remedy.”

21. Discussion: Were the data missing at random? Were those with complete data different than those with incomplete data?

*****There were systematic differences between patients with with complete and with incomplete data. These differences were taken into account by the chosen imputation model. 

The following statements are included in the limitation section:

- …. Second, there is a selection bias due to the incomplete follow up. Non-responding study participants were younger, less educated, more often smokers, and less frequently employed. At discharge from CR, they reported a lower health status. These differences were taken into account by the imputation model. However, there may be reasons for missingness that were not covered by our data. This has to be kept in mind when interpreting the study results. 

- … Patients were allowed to refrain from answering parts of the questionnaires or individual questions. Missing values were imputed in the statistical models to balance the data. However, it may be that these modeling approaches do not compensate the selection effects.

22. Reference 1 and 35 are the same.

*****Thank you. We corrected this mistake.

Reviewer #1

This paper has hallmarks of good clinical and ecological validity in light of its links with a patient outcome registry. It would be helpful if the Abstract results included actual data, as opposed to statements which are more like conclusions

*****Thank you very much. We revised the abstract and present now a selection of the most important findings. 

Whilst desire to retire was an interesting and important covariate to RTW, would not age also be key, in respecting the Methods clearly had a cut-off of 65 years.

*****Age was included as potential covariate for each of the endpoints, but was only selected for explanation of MCS. 

There are a number of risk factor data reported and mention of the Borg scale. However, their relevance to all this isn't clear. Furthermore, the changes in most risk factors, were these actually related to the Rehab process or merely functions of medications prescribed.

*****There are several parameters including sociodemographic, rehabilitation indication, comorbidities and cardiovascular risk factors, which are predominantly reported to describe the investigated patient population. This is a prerequisite to ensure transparency regarding the comparability of our population to others of interest. However, relevant parameters, in particular modifiable risk factors, were taken into account in the statistic modeling. The statistic section was modified to fully describe the modeling process. 

The risk factor management including the prescription of medications for secondary prevention is an essential part of the German standardized CR program. The effect of this program on risk factors cannot be differentiated for single program components.

The BORG rating is irrelevant to the presented results – we removed this information from the text.

Line 105 mentions attendance of 12 training units per week? What is a training unit? In terms of programme fidelity and in addition to programme uptake and completion, how compliant were the participants with attending? (e.g. participation rates per week).

*****A training unit consists of up to 30 Minutes endurance or resistance training or gymnastics, or up to 45 Minutes outdoor walking. The duration and intensity depends on the assigned training group and physical performance of the individual patient. The CR program was conducted mostly as an inpatient intervention with mandatory all-day activities, which were specified for the patient. We did not capture patients’ compliance, but we generally assume that it was high due to the obligatory character of the intervention. We extended the description of the conducted rehab program as provided by the German pension insurance and hope for better understanding: 

“All patients performed a comprehensive standardised CR programme with a regular duration of three to four weeks according to the specifications of the German pension insurance.[16] The programme can be performed in either an inpatient or outpatient setting and consists of all-day activities including counseling by a cardiologist, risk-factor modification strategies (e.g. patient education on nutritional habits, smoking cessation, physical activity, and medication adherence), physician-supervised exercise training and sports therapy (e.g. training on a bicycle ergometer, outdoor walking, resistance training, gymnastics), psychosocial interventions (health education and counseling, psychotherapy, stress management in single or group sessions), and vocational assessment and physician and social worker counseling. [17,18] On average, patients perform 12 training and sports therapy units per week with a duration –depending on the training group and physical performance – up to 30 Minutes and 45 Minutes for outdoor walking, respectively, and 8 additional counselling sessions.[19]”

The Exercise Capacity was measured but not factored into the relationship with RTW? A way to check on whether exercise capacity had changed was to see if patients heart rate or RPE was lower for any give workmate in Watts (submit or max), which in fact you have reported

*****The change of exercise capacity is considered in the variable ‘endurance training load [watt]’. This changes during CR achieved no significance for the final model, while the endurance training load at admission to CR turned out to be a predictor of RTW.

More clarity required whether RTW data included those already unemployed and or on sick leave prior to commencement of CR (apologies for my oversight if this has been stated)

*****At baseline, 1127 patients (90.3%) were employed, 94 (7.5%) were unemployed (others were homemaker or in temporary annuity). 892 patients (71.5%) were on sick leave before CR (table 1). RTW was analysed for all enrolled patients. We added this information into the section “Statistics”: 

“For all further analyses, only patients with a non-missing outcome in the original dataset were included, regardless of the employment status at baseline.” (line 201-203)

There seem to be lots of abbreviations; can these be reduced? or at least make sure they are clearly defined in legends for all Tables and Figures

*****The most abbreviations result from the use of several questionnaires. We revised all table and figure legends and the manuscript regarding the consistent use of abbreviations. Apart from the abstract, we also omit the abbreviation of return-to-work and patient reported outcome measures in the manuscript. We hope it is easier to read now. 

Reviewer #2

This paper aimed to identify predictors of return to work (RTW) and quality of life (QOL) 6 months after cardiac rehabilitation. Although the paper describes an interesting topic, I do have quit some concerns and comments that need attention. Some major revisions need to be made in the manuscript before this manuscript can be reviewed in detail.

Major comments:

1. The authors should be more clear what is the novelty of their paper. In the introduction the authors describe that several factors are known to influence RTW (such as depression, anxiety, expectations) in lines 69-72. However, in line 77 it is said that few is known about the association between patient centered outcomes and RTW. Are the factors mentioned in lines 69-72 not also patient centered outcomes? And in line 62, it is described that the effect of CR on QOL is well-known, so why is QOL choses as an outcome in this paper?

*****The introduction was extensively revised. We hope for a better understanding of our intention. 

2. The methods section describing the study outcomes/ predictors is extremely difficult to follow (lines 116-136), partly due to the large amount of parameters. Could the authors try to re-organize this paragraph and make clear what exactly is measured and which instrument is used (for instance with a table or using bullet points).

*****According to your suggestion, we re-organized the paragraph using bullet points, hoping it is now a more structured concept.

3. Could the authors give some more information on the content of the CR program (duration, training sessions and counselling sessions). Twelve training sessions per week seem quite high.

*****We extended the description of the conducted rehab program as provided by the German pension insurance and hope, that it is clearer now:

“All patients performed a comprehensive standardised CR programme with a regular duration of three to four weeks according to the specifications of the German pension insurance.[16] The programme can be performed in either an inpatient or outpatient setting and consists of all-day activities including counseling by a cardiologist, risk-factor modification strategies (e.g. patient education on nutritional habits, smoking cessation, physical activity, and medication adherence), physician-supervised exercise training and sports therapy (e.g. training on a bicycle ergometer, outdoor walking, resistance training, gymnastics), psychosocial interventions (health education and counseling, psychotherapy, stress management in single or group sessions), and vocational assessment and physician and social worker counseling. [17,18] On average, patients perform 12 training and sports therapy units per week with a duration –depending on the training group and physical performance – up to 30 Minutes and 45 Minutes for outdoor walking, respectively, and 8 additional counselling sessions.[19]”

4. With regard to the used statistics I have some questions:

a. Were all predictors added to the same multiple linear regression model? I can imagine that there might be some overlap and correlation between some of the predictors. Did the authors test for multicollinearity?

*****We studied potential multicollinarity by calculation of the variance inflation factors (VIF) for the complete set of potential regressors. VIF were distinctly below the generally accepted limit of 10 with one exception: mental health at baseline as assessed by use of the IRES-24 had an VIF of 10.48 which means that the corresponding coefficients might be slightly compromised by multicollinearity. We added to the limitations section: “Multicollinearity may limit the interpretability of model coefficients. However, the variance inflation factors of the studied predictors were distinctly below the generally accepted limit of 10 with one exception: the mental health at admission had a variance inflation factor of 10.48. Altogether, the multivariate models were sufficiently stable to allow clinical interpretations.” 

b. Could the authors add some information about the power of their study to test for this large amount of predictors?

*****We assume that despite the lack of multicollinearity the models with the full set of covariates (which are now shown in the appendix) are overfitted which results in a lack of power. However, after variable selection the models were robust and performed quite well to identify several predictors of return to work and HRQL

c. Why was chosen for only one imputation and not for multiple imputations?

*****Thank you for this valuable comment. We recalculated all models and performed the backward selections with 20 imputations instead of one as described in detail in the statistics paragraph. 

5. With regard to generalizability of the data: could the authors say something about patients that were not willing to participate in this study? Is the sample that was willing to participate representative of the whole population?

*****Patients without an informed consent were excluded from this study. Due to privacy, it was not allowed to capture data for these patients. However, our participants were similar to our target population (cardiac patients ≤65 years of age) according to statistics on the rehabilitation services of the German pension insurance with regard to age and sex. We added this information including a reference to the limitations (line 390-392).

6. Please clarify in your paper what is seen as a PROM and what is seen as subjective health (and what is the difference between these two concepts).

*****‚Subjective health‘ is a key area of cardiac rehabilitation (line 106) meaning that the interprofessional CR team strives to improve the perceived health status of a patient in a holistic treatment approach. We used PROMs to operationalize subjective health, as they are “questionnaires completed by patients to assess the effects of disease or treatment (or both) on symptoms, functioning, and health-related quality of life from their perspective (Calvert M et al. 2019). In the methods section, we stated now: “… The majority of captured data were taken from patients’ records, while social medicine and subjective health were assessed by means of specific patient-reported outcomes measures used in the study: …” (line 139-141).

7. In the paper there is a lot of attention focused on the role of pension desire in RTW. What is the rationale behind investigating this? If patients have the desire and possibility to retire, should that be seen as a negative outcome of CR? In my opinion, it should not be seen like that. It can be a good and well thought out choice. Reading your results, one could conclude that to improve RTW we should try to change someone’s wish to retire during CR. What is the vision of the authors on this?

*****As described in the methods section, this paper is part of the OutCaRe study, that started with a Delphi expert survey of 70 cardiologists and other professionals. These experts extracted and consented potential indicators of rehabilitation success. All chosen parameters including pension desire were analyzed in terms of their predictive value for return-to-work and HRQL after CR according to the predefined methodological approach. In this analysis, pension desire as it was one of only two parameters, that predict both return-to-work and HRQL (discussion, first paragraph). For this reason, pension desire should be taken into account for any predictive modelling of return to work, at least as a control variable.

Pension desire is an item of the Würzburger Screening, a valid instrument identifying occupational issues and the need for vocational rehabilitation. 

However, patients’ pension desire is not synonymous with the possibility to retire. Retirement before pension age requires a proven permanent significant limitation of occupational capacity. Patients in our study with pension desire at admission to CR were retired in only 16%, 33% were at sick leave, while 37% had returned to work within a half year after CR. Pension desire may be a surrogate parameter, that implicates aspects of non-captured occupation-related factors as well as disease-related aspects (see discussion section). Indeed, the occupational resumption is the main goal of CR for the German pension insurance, which is the funding institution for the most CR patients before the retirement age of 65

8. The authors chose to look for predictors of successful return to work. On top of patients still on sick leave, the group of patient with unsuccessful return to work included patients that chose to retire, patient that applied for pension and patient that were already unemployed before CR. Wouldn’t it be more interesting to look for predictors of patients that failed to return to work as a result of their cardiac disease, even though they would have wanted to (the group of 15% still on sick leave)? This would be the group that might benefit of extra support during or after CR.

*****Since successful return to work is a main goal of CR by law for all patients of employable age, it was per protocol the predefined aim of this study to systematically investigate a multitude of modifiable clinical parameters and patient-reported outcome measures regarding their impact on return-to-work. Of course, the group still on sick leave is of special interest. However, this group comprises only 190 patients. In the study, in total 1586 patients were enrolled, 1262 patients responded to the follow-up-survey. Thus, it is a high selected patient population. Results of such an analysis would be difficult to translate into clinical practice because patients who are on sick leave at follow up cannot be identified at CR admission.

9. The percentages in lines 215-217 add up to 102.3%.

*****Multiple answers were possible. Patients could be employed or unemployed or at sick leave if they had applied for pension or could be unemployed and at sick leave.

10. Lines 228-251 in the results are difficult to follow. Could the authors clarify if the mentioned predictors are measured at start or completion of CR or concern an improvement during CR?

*****We revised the regarding paragraphs and included consistently the respective capturing times of the parameters.

11. If I understand well, figures 2 and 3 only show outcomes of significant results (predictors). This gives a misleading representation of all results. I would suggest to add all variables that were tested to this figure so that it is clear which predictors were significant and which were not.

*****The models with the full set of covariates are now shown in the supplement (figures S1-S3). However, these models are not very powerful and thus not appropriate for identification of relevant predictors. We thus preferred to keep the results of the models with variable selection in the main part of the paper.

12. The discussion is quite difficult to follow. When are results of other authors discussed and when does it concern the author’s own results? Also, it is sometimes difficult to follow if the outcomes with regard to QOL or RTW are discussed. And how can these results be used in daily clinical practice. I suggest a re-organization of this section (first discuss own results (split RTW and QOL), second how does this compare to results of other studies, third what can be done differently with this information in daily CR practice) and a “deeper” discussion of the work.

*****We are grateful for this suggestion. For easier readability, we rewrote the discussion. We start with the description of main results, followed by predictors of RTW, HRQL and both with a deeper discussion of the latter, subsequently discussing non-intuitive results. Finally, we discuss the impact of modifications in PROMs during CR and the usefulness of an interprofessional team and finish with the limitations of the study.

Smaller remarks:

1. Please try to be consistent in your manuscript with the use of abbreviations.

*****We revised the manuscript regarding the consistent use of abbreviations.

2. What do the authors mean with “mid-term associations”?

*****We meant associations … in the mid-term course after CR. We rephrased this sentence accordingly (line 91).

3. Could the authors try to rephrase the aim in the abstract? It is a complex sentence that I had to read several times to understand.

*****We extensively revised the abstract as well as the the aim of our investigation: “… Out of a multitude of variables collected at CR admission and discharge, we aimed to identify predictors of returning to work (RTW) and HRQL 6 months after CR.” 

4. What do the authors mean with occupational study? (line 29)

*****We corrected this mistake. The correct term is “observational”

5. In Table 1 :

a. living situation does not add up to 100%.

*****The category „others“ with 3.4% was missing (e.g. commune). We added it into the table 1.

b. What is meant with the number 3.4 +/- 1.6 behind comorbidities?

*****This is the mean and standard deviation of number of comorbidities. We added the information into the table legend.

c. Is AF a diagnosis or comorbidity?

*****AF (atrial fibrillation) is a comorbidity, see table 1.

6. In Table 2: could the author add p-values?

*****Yes, indeed. We added a column with the p-values.

7. Social integration is not the same as return to work (it also includes return to leisure time activities etc). I would suggest to not mix these terms.

*****Thank you for this hint. We substituted ‘social integration’ by ‘occupational resumption’.

8. How many centres were involved in this study?

*****12 rehabilitation centers were involved. Please see line 112.

9. The description of the figures has a lot of overlap with the text.

*****We removed the redundant text from the figure legends.

---

## [Decision Letter · Decision Letter 1]

14 Feb 2020

PONE-D-19-29345R1

Patient-reported outcomes predict return to work and health-related quality of life 6 months after cardiac rehabilitation: Results from a German multi-centre registry (OutCaRe)

PLOS ONE

Dear Dr. Salzwedel,

Thank you for submitting your manuscript to PLOS ONE. After careful consideration, we feel that it has merit but does not fully meet PLOS ONE’s publication criteria as it currently stands. Therefore, we invite you to submit a revised version of the manuscript that addresses the points raised during the review process.

We would appreciate receiving your revised manuscript by Mar 30 2020 11:59PM. To enhance the reproducibility of your results, we recommend that if applicable you deposit your laboratory protocols in protocols.io, where a protocol can be assigned its own identifier (DOI) such that it can be cited independently in the future. For instructions see: http://journals.plos.org/plosone/s/submission-guidelines#loc-laboratory-protocols

We look forward to receiving your revised manuscript.

Kind regards,

Stephanie Prince Ware, PhD

Academic Editor

PLOS ONE

Additional Editor Comments (if provided):

I thank the authors for their detailed consideration of the points previously raised by myself and the two reviewers. Both original reviewers re-assessed the manuscript and have considerations that remain in order for this manuscript to be considered for publication. Please find their feedback and some points raised by myself below:

-See my previous comment #10 concerning the RTW analysis: I am still not completely clear why those who were not working at baseline were not excluded.

-Previous comment #15 concerning figures 2 and 3: in their description please list all variables in the models.

Reviewers' comments:

Reviewer's Responses to Questions

**Comments to the Author**

1. If the authors have adequately addressed your comments raised in a previous round of review and you feel that this manuscript is now acceptable for publication, you may indicate that here to bypass the “Comments to the Author” section, enter your conflict of interest statement in the “Confidential to Editor” section, and submit your "Accept" recommendation.

Reviewer #1: All comments have been addressed

Reviewer #2: (No Response)

2. Is the manuscript technically sound, and do the data support the conclusions?

Reviewer #1: Yes

Reviewer #2: Yes

3. Has the statistical analysis been performed appropriately and rigorously? 

Reviewer #1: I Don't Know

Reviewer #2: Yes

4. Have the authors made all data underlying the findings in their manuscript fully available?

Reviewer #1: Yes

Reviewer #2: Yes

5. Is the manuscript presented in an intelligible fashion and written in standard English?

Reviewer #1: Yes

Reviewer #2: Yes

6. Review Comments to the Author

Reviewer #1: My one main concern continues to be that since this included a strong for on an exercise-based intervention, the authors still seem to be confident on their results on changes (or in fact no change in functional capacity) purely based on the submaxmial "exercise training dose" not significantly being changed over time. This represents only a psychological desire for whatever the reason is to who and how the intensity should have been progressed over time as a very fundamental element of any exercise training programme. Unless there is a change in some measured physiological parameter like a reduction in HR for a given HR or other physiological changes demonstrating a change, the authors can not be certain that no change has occurred.

Reviewer #2: I would like to compliment the authors on their revisions. The document has greatly improved. Nevertheless, I still have some remarks that I hope the authors are willing to take into consideration. I mainly feel that the discussion section could be further improved.

Major remarks

1. The introduction greatly improved as compared to the previous version. Nevertheless, I am still not convinced of the added value of investigating predictors of HRQL. The introduction states that there are mainly inconclusive results with regard to return to work with suboptimal outcomes after CR. Lines 85-88 seem to suggest that predictors for HRQL are already known. In line 73 the authors state that return to work influences HRQL, but why do the authors decide to separately investigate predictors of return to work AND HRQL and not just return to work?

2. A clear research aim/goal is missing in the introduction

3. In the statistics sessions the MID is mentioned for part of the patient-reported outcomes, but not for all. How did you deal with the other outcomes, such as the WHO-5, ASKU and Wurzburger list?

4. The discussion sessions has greatly improved with your extra information, but is now quit long. I suggest to shorten this section and also rearrange a bit the paragraphs. Start with discussing the predictors that you found and how you think (based on literature) these predictors could influence RTW/QOL (this last information is for instance missing for heart-focused anxiety). Move the paragraph lines 349-358 to the end and combine this paragraph with lines 382-389. Describe in this paragraph not only what is NOT working for RTW & HRQL, but also give clear suggestions (based on your outcomes) what should be done to improve CR.

5. There seems to be some grammar errors in the paper. I recommend consulting a native speaker/ professional to check the paper.

Smaller remarks

1. In the abstract ‘lifestyle change motivation’ is defined as risk factor management, shouldn’t this be part of patient-reported outcomes?

2. I advise to split lines 42-46 (abstract) in two sentences to increase readability

3. Please make clear in your abstract what predictors were measured at discharge and which predictor concern changes during CR.

4. Could you add a reference to line 73 that return to work influences HRQL?

5. Line 74-77 it is unclear whether outcomes or predictors are discussed (e.g. functional status is a predictors and HRQL an outcome?). I advise to delete this sentence.

6. Please also add to your methods section that you check for multicollinearity.

7. Sex and gender are both used in your document. Please pick one (I prefer sex).

8. I would recommend adding a subheading “Predictors” to page 6 to increase readability.

9. Nice summary at the start of the discussion. Could you shortly add (one sentence) how RTW and QOL were improved at 6 months follow-up?

7. PLOS authors have the option to publish the peer review history of their article (what does this mean?). If published, this will include your full peer review and any attached files.

Reviewer #1: No

Reviewer #2: No

---

## [Author Response · Author response to Decision Letter 1]

19 Mar 2020

We thank the editor and the reviewers for re-reviewing our manuscript. Please find our responses to the comments below. Changes in the manuscript text are highlighted in grey. 

Sincerely,

Annett Salzwedel

for the authors of the manuscript.

Editor:

-See my previous comment #10 concerning the RTW analysis: I am still not completely clear why those who were not working at baseline were not excluded.

>>>A large percentage of those undergoing rehabilitation generally work, but have been on sick leave since their cardiac event and the corresponding hospitalization until rehabilitation and beyond. Cardiac rehabilitation in Germany has the mission per law to support RTW for patients in employable age even if they were at sick leave or unemployed before rehab. Our choosen method represents this real world goal of rehabilitation in Germany. We tried to clarify this fact in the methods: Return-to-work was analyzed regardless of the employment status at baseline as it is a goal of CR in Germany to support return-to-work even in patients at sick leave or unemployed before CR. (line 201-203)

-Previous comment #15 concerning figures 2 and 3: in their description please list all variables in the models.

>>>As requested, we now listed all variables in the description of the figures as follows:

The forest plot shows the final model after backward selection. The following variables were taken into account in the starting model (see supplemental material, figure S1-S3): at admission to rehabilitation: sex, smoking, lifestyle change motivation, pension desire, self-assessed occupational prognosis, self-assessed health prognosis, living situation, educational level, rehabilitation referral for chronic disorder, comorbidities (diabetes mellitus, depression, peripheral artery disease, heart failure, diseases of the back and spine, stress at work, stress by major life events, age, systolic/diastolic blood pressure, body mass index, endurance training load, depression (PHQ-9), heart-focused anxiety (HAF-17), well-being (WHO-5), physical/mental component summary in the SF-12, physical/mental health and pain in the IRES-24, self-efficacy (ASKU); at discharge from rehabiliation: Sm smoking, lifestyle change motivation, pension desire, self-assessed occupational prognosis, self-assessed health prognosis; changes during rehabilitation: systolic/diastolic blood pressure, body mass index, endurance training load, depression (PHQ-9), heart-focused anxiety (HAF-17), well-being (WHO-5), physical/mental component summary in the SF-12, physical/mental health and pain in the IRES-24, self-efficacy (ASKU).

Reviewer #1: My one main concern continues to be that since this included a strong for on an exercise-based intervention, the authors still seem to be confident on their results on changes (or in fact no change in functional capacity) purely based on the submaxmial "exercise training dose" not significantly being changed over time. This represents only a psychological desire for whatever the reason is to who and how the intensity should have been progressed over time as a very fundamental element of any exercise training programme. Unless there is a change in some measured physiological parameter like a reduction in HR for a given HR or other physiological changes demonstrating a change, the authors can not be certain that no change has occurred.

>>>In fact, the measured parameters of physical performance (maximum exercise capacity in the bycicle ergometry stress test, endurance training load and 6-min walking distance) showed the largest enhancements during cardiac rehabilitation with p-values <0.001 and standardized effect sizes up to >1 (see table 2) compared with the other measured parameters. Therefore, in our opinion, these parameters of physical performance suggested by experts as indicators of rehabilitation success were adequately confirmed by our study. Moreover, the changes in the endurance training load increase the probability of return-to-work statistically significantly, but to a small extend (OR 1.09 per 5 Watts). The other parameters of physical performance were not taken into account in the multivariable modeling due to the content overlap and less available data.

Reviewer #2: I would like to compliment the authors on their revisions. The document has greatly improved. Nevertheless, I still have some remarks that I hope the authors are willing to take into consideration. I mainly feel that the discussion section could be further improved.

Major remarks

1. The introduction greatly improved as compared to the previous version. Nevertheless, I am still not convinced of the added value of investigating predictors of HRQL. The introduction states that there are mainly inconclusive results with regard to return to work with suboptimal outcomes after CR. Lines 85-88 seem to suggest that predictors for HRQL are already known. In line 73 the authors state that return to work influences HRQL, but why do the authors decide to separately investigate predictors of return to work AND HRQL and not just return to work?

>>>The relationship between return-to-work and HRQL is probably bi-directional and not causal in only one direction. There are some studies investigating predictors for occupational reintegration and HRQL, but usually not in the same study with the same set of parameters for both outcomes. Moreover, the multitude of parameters that can be modified by CR and in particular patient reported outcomes was not systematically taken into account. We added the following connecting reasoning in line 87-88: Since return-to-work and HRQL after a cardiac event are closely associated, [12] several common predictors can be assumed. (line 87-88)

2. A clear research aim/goal is missing in the introduction

>>>We added the following aim: In the OutCaRe-study we aimed to identify predictors of occupational reintegration and HRQL among the same set of patient-reported outcome measures, clinical parameters, cardiovascular risk factors and physical performance. (line 92-94)

3. In the statistics sessions the MID is mentioned for part of the patient-reported outcomes, but not for all. How did you deal with the other outcomes, such as the WHO-5, ASKU and Wurzburger list?

>>>The use of MID was a complementary method (in addition to statistical significance and determination of effect size) for evaluating changes in patient reported outcomes, which was added as requested by the editor. MIDs are not available for all measures. We modified the regarding sentence in the methods section as follows: In addition, the changes in patient-reported outcome measures (metric variables) during CR were assessed using the minimal important difference (MID) if available. (line 190-191) The MID of WHO-5 is 10% as stated below (line 193-194). Statistical significance and standardized effect size were used for the evaluation of ASKU (no MID available). The Wurzburger screening generates categorical variables, no MID exists.

4. The discussion sessions has greatly improved with your extra information, but is now quit long. I suggest to shorten this section and also rearrange a bit the paragraphs. Start with discussing the predictors that you found and how you think (based on literature) these predictors could influence RTW/QOL (this last information is for instance missing for heart-focused anxiety). Move the paragraph lines 349-358 to the end and combine this paragraph with lines 382-389. Describe in this paragraph not only what is NOT working for RTW & HRQL, but also give clear suggestions (based on your outcomes) what should be done to improve CR.

>>>Thank you for your suggestion. In the first revision, we added several items to the discussion section as requested by the editor and reviewers (e.g. MID, educational level and return to work, HRQL related predictors, multicollinearity, benefit of social workers, missing data). Taking into account these several issues, we rearranged and rewrote the discussion section.

From our point of view, the paragraph discussing the benefit of social workers in CR is placed in a logical order to the paragraph above. Moving the paragraph would take the content out of the context.

In the last paragraph above the limitation section we recommend to provide multiprofessional CR programms including cardiologists, social workers and sports therapists or physiotherapists on the basis of an individualized treatment approach. For this we recommend the consistent implementation of questionnaires assessing subjective health (see conclusion). More recommendations are not supported by our data.

5. There seems to be some grammar errors in the paper. I recommend consulting a native speaker/ professional to check the paper.

>>>The paper has been revised by a professional proof reader.

 

Smaller remarks 

>>>We considered all following remarks.

1. In the abstract ‘lifestyle change motivation’ is defined as risk factor management, shouldn’t this be part of patient-reported outcomes?

2. I advise to split lines 42-46 (abstract) in two sentences to increase readability

3. Please make clear in your abstract what predictors were measured at discharge and which predictor concern changes during CR.

4. Could you add a reference to line 73 that return to work influences HRQL?

5. Line 74-77 it is unclear whether outcomes or predictors are discussed (e.g. functional status is a predictors and HRQL an outcome?). I advise to delete this sentence.

6. Please also add to your methods section that you check for multicollinearity.

7. Sex and gender are both used in your document. Please pick one (I prefer sex).

8. I would recommend adding a subheading “Predictors” to page 6 to increase readability.

9. Nice summary at the start of the discussion. Could you shortly add (one sentence) how RTW and QOL were improved at 6 months follow-up?

---

## [Editor Report · Decision Letter 2]

20 Mar 2020

PONE-D-19-29345R2

Patient-reported outcomes predict return to work and health-related quality of life 6 months after cardiac rehabilitation: Results from a German multi-centre registry (OutCaRe)

PLOS ONE

Dear Dr. Salzwedel,

Thank you for submitting your manuscript to PLOS ONE. After careful consideration, we feel that it has merit but does not fully meet PLOS ONE’s publication criteria as it currently stands. Therefore, we invite you to submit a revised version of the manuscript that addresses the points raised during the review process.

Thank you for addressing the comments by the reviewers. I feel that you have adequately addressed them all. My remaining issues are very minor and mostly grammar related. Please make these final changes as described below.

We would appreciate receiving your revised manuscript by May 04 2020 11:59PM. To enhance the reproducibility of your results, we recommend that if applicable you deposit your laboratory protocols in protocols.io, where a protocol can be assigned its own identifier (DOI) such that it can be cited independently in the future. For instructions see: http://journals.plos.org/plosone/s/submission-guidelines#loc-laboratory-protocols

We look forward to receiving your revised manuscript.

Kind regards,

Stephanie Prince Ware, PhD

Academic Editor

PLOS ONE

Additional Editor Comments (if provided):

Abstract, line 27, please insert a comma before "as well as".

In the title and throughout the text, please write numbers under 10 using letters (e.g., 6 = six)

Abstract, please include '=' signs after OR and colons (:) after CI.

Abstract, line 52: please change this to "patients with heart disease".

Line 75, remove "the" at the end of this line.

Line 85, there appears to be a comma missing between exercise and coronary?

Throughout the text please change to patients living with heart disease vs. heart disease patients.

Line 91, insert a comma before "as well as".

Line 91, insert in brackets after mid-term course what you mean by this (e.g., provide time frame)

Line 92, define OutCaRe-study upon first occurrence.

Line 97, insert a comma before "as well as".

Lines 105-108: please place these items on the same line and use semicolons (;) to separate them.

Lines 129-130: please use commas instead of - to separate the training group indication. i.e. with a duration, depending on the training group and physical performance, of up to...

Line 218, define ACS

Lines 219-221: explain your cut-off used for VIFs

Line 235: Write out 18 (i.e. Eighteen)

Line 275: Write out 89 (i.e. Eighty nine)

Line 278-282: this sentence is too long. Consider separating into two sentences the first ending after "summary scale" on line 280 and the second beginning with "Scores for HRQL six months after CR..."

Line 281 and throughout manuscript, include = after OR

Line 292: consider revising writing to: "mental-HRQL"

Line 356, insert a comma before "as well as".

Line 433, change "whole population" to the general CR population

Line 459, insert a comma before "as well as".

Please be consistent in your writing of "return to work" vs. "return-to-work"

Did the authors check if their non-imputed models achieved similar results as the imputed models?
---

## [Author Response · Author response to Decision Letter 2]

20 Apr 2020

We thank the editor and the reviewers for re-reviewing our manuscript. Please find our responses to the comments below. Changes in the manuscript text are highlighted in grey. 

Sincerely,

Annett Salzwedel & Karl Wegscheider

for the authors of the manuscript.

Editor:

****Thank you very much for the hints, which we’ve all considered.

Abstract, line 27, please insert a comma before "as well as".

In the title and throughout the text, please write numbers under 10 using letters (e.g., 6 = six)

Abstract, please include '=' signs after OR and colons (:) after CI.

Abstract, line 52: please change this to "patients with heart disease".

Line 75, remove "the" at the end of this line.

Line 85, there appears to be a comma missing between exercise and coronary?

Throughout the text please change to patients living with heart disease vs. heart disease patients.

Line 91, insert a comma before "as well as".

Line 91, insert in brackets after mid-term course what you mean by this (e.g., provide time frame)

Line 92, define OutCaRe-study upon first occurrence.

Line 97, insert a comma before "as well as".

Lines 105-108: please place these items on the same line and use semicolons (;) to separate them.

Lines 129-130: please use commas instead of - to separate the training group indication. i.e. with a duration, depending on the training group and physical performance, of up to...

Line 218, define ACS

****All done.

Lines 219-221: explain your cut-off used for VIFs

****We added the following explanation to the statistics: For the complete set of potential regressors, potential multicollinarity was studied by calculating the variance inflation factors, values below 10 were considered acceptable. (line 217-218)

Line 235: Write out 18 (i.e. Eighteen)

Line 275: Write out 89 (i.e. Eighty nine)

Line 278-282: this sentence is too long. Consider separating into two sentences the first ending after "summary scale" on line 280 and the second beginning with "Scores for HRQL six months after CR..."

Line 281 and throughout manuscript, include = after OR

Line 292: consider revising writing to: "mental-HRQL"

Line 356, insert a comma before "as well as".

Line 433, change "whole population" to the general CR population

Line 459, insert a comma before "as well as".

Please be consistent in your writing of "return to work" vs. "return-to-work"

****All done.

Did the authors check if their non-imputed models achieved similar results as the imputed models?

****Yes, actually we started with a non-imputed model. However, the type of model that is adequate for the problem at hand require complete data sets. Without imputation, the analysis has to be done in the markedly smaller data set of complete cases (915 (73%) of 1262 cases) which may not be representative for the whole study population. Correspondingly, the analysis of the non-imputed data resulted in another set of selected variables that makes it difficult to compare the resulting models. In such cases, the imputed model is usually thought to be more reliable than the complete case model.

---

## [Editor Report · Decision Letter 3]

22 Apr 2020

Patient-reported outcomes predict return to work and health-related quality of life six months after cardiac rehabilitation: Results from a German multi-centre registry (OutCaRe)

PONE-D-19-29345R3

Dear Dr. Salzwedel,

We are pleased to inform you that your manuscript has been judged scientifically suitable for publication and will be formally accepted for publication once it complies with all outstanding technical requirements.

With kind regards,

Stephanie Prince Ware, PhD

Academic Editor

PLOS ONE
---

## [Editor Report · Acceptance letter]

24 Apr 2020

PONE-D-19-29345R3 

Patient-reported outcomes predict return to work and health-related quality of life six months after cardiac rehabilitation: Results from a German multi-centre registry (OutCaRe) 

Dear Dr. Salzwedel:

I am pleased to inform you that your manuscript has been deemed suitable for publication in PLOS ONE. Congratulations! Your manuscript is now with our production department. 

With kind regards,

on behalf of

Dr. Stephanie Prince Ware 

Academic Editor

PLOS ONE